

# Characteristics of the tropical tropopause inversion layer using high-resolution temperature profiles retrieved from COSMIC GNSS Radio Occultation

Noersomadi Noersomadi[1,2], Toshitaka Tsuda[1], Masatomo Fujiwara[3]

[1]Research Institute for Sustainable Humanosphere (RISH), Kyoto University, Uji, 611-0011, Japan
[2]National Institute of Aeronautics and Space (LAPAN), Bandung, 40173, Indonesia
[3]Faculty of Environmental Earth Science, Hokkaido University, Sapporo, 060-0810, Japan

*Correspondence to*: Noersomadi (noersomadi@rish.kyoto-u.ac.jp)

**Abstract.** Using COSMIC GNSS Radio Occultation (RO) observations from January 2007 to December 2016, we retrieved temperature profiles with 0.1 km vertical resolution in the upper troposphere and lower stratosphere (UTLS). We investigated the global distribution of static stability ($N^2$) and the characteristics of the tropopause inversion layer (TIL) in the tropics, where a large change in temperature gradient occurs associated with sharp variations of $N^2$. We show the variations of the mean $N^2$ profiles in conventional height coordinates as well as in coordinates relative to both the Lapse Rate Tropopause (LRT) and the Cold Point Tropopause (CPT). When the $N^2$ profiles are averaged relative to CPT height, there is a very thin (<1 km) layer with average maximum $N^2$ in the range $11.0–12.0 \times 10^{-4}$ s$^{-2}$. The mean and standard deviation of the tropopause sharpness ($S$-ab), defined as the difference between the maximum $N^2$ ($maxN^2$) and minimum $N^2$ ($minN^2$) within ±1 km of the CPT, is $(10.5 \pm 3.7) \times 10^{-4}$ s$^{-2}$. About 70% of the values of TIL thickness ($dH$), which is the thickness of the layer over which $N^2 \geq 0.8maxN^2$, were in the range $0.4 \pm 0.04$ km.

We focused on the variation of $S$-ab in two longitude regions, 90°–150°E (Maritime Continent; MC) and 170°–230°E (Pacific Ocean; PO), with different land–sea distribution. Seasonal variations of $S$-ab and $dH$ were related to the deep convective activity represented by low Outgoing Longwave Radiation (OLR) during the Australian and Asian monsoons. The $S$-ab anomaly ($S$-ab*) was out-of-phase with the OLR anomaly in both the MC and PO. The correlation between $S$-ab* over the MC and PO and the sea surface temperature (SST) Niño 3.4 index was –0.66 and +0.88, respectively. This means that during La Niña (SST Nino 3.4 < –0.5 K) in the MC, and El Niño (SST Nino 3.4 > +0.5 K) in the PO, warmer SSTs in the MC and PO produce more active deep convection that tends to force the air upward to the tropopause layer and increase the temperature gradient there. The intra-seasonal variation in $S$-ab* during slow and fast episodes of the Madden–Julian Oscillation (MJO) demonstrates that eastward propagation of positive $S$-ab* is associated with organized deep convection. This suggests that convective activity in the tropics is a major control on variations in tropopause sharpness at intra-seasonal to interannual time-scales.

Keywords: COSMIC GNSS-RO, tropopause inversion layer, tropics



## 1 Introduction

The tropical tropopause layer (TTL) at a height of 12–19 km plays an important role in the Earth's climate, as tropospheric air enters the stratosphere mainly in this region, where Stratosphere–Troposphere Exchange (STE) processes occur (Holton et al., 1995). The dynamics and radiative processes in the TTL have received much attention in the decades since the first observation of a slow meridional circulation, the Brewer–Dobson circulation (e.g., Butchart, 2014). In the tropics, the tropopause is defined as either the lapse rate tropopause (LRT) or the cold point tropopause (CPT). Here, the LRT refers to "the lowest level where the temperature lapse rate decreases to 2 K/km provided the average between this and higher levels within 2 km does not exceed 2 K/km" (World Meteorological Organization; WMO, 1957), while the CPT is the level of the minimum temperature. The tropopause inversion layer (TIL) is a narrow layer about 1–2 km from the CPT that is characterized by a sharp change in the vertical gradient of the temperature profile, which is also recognized in the static stability profile (Bell and Geller, 2008; Birner et al., 2002). It is a boundary within the TTL that controls the mixing of air between the troposphere and stratosphere (e.g. Fujiwara and Takahashi, 2001). A very low temperature in the TIL causes dehydration of the air entering the stratosphere (Mote et al., 1996; Fueglistaler et al., 2009).

The static stability is expressed as the square of the Brunt Väisälä or buoyancy frequency ($N^2$). $N^2$ is the main factor in the dispersion relations for atmospheric waves, including medium-scale gravity waves and planetary-scale equatorial waves such as Kelvin waves and mixed Rossby–gravity waves (Andrew et al., 1987). The spectrum of normalized temperature fluctuations associated with gravity waves is sensitive to change in $N^2$ (Smith et al., 1987; Fritts et al., 1988). Therefore, an accurate understanding of its horizontal distribution and temporal variations, particularly in the tropics, is important when investigating the characteristics of atmospheric wave propagation. The 20-day Kelvin wave propagation influences the structure of the tropopause height and the value of $N^2$ (Tsuda et al., 1994). A long-term analysis using Singapore radiosonde data showed that Kelvin wave activity was stronger during the transition from the easterly to westerly phase of the Quasi Biennial Oscillation (QBO) (Shiotani and Horinouchi, 1993; Randel et al., 2003; Ratnam et al., 2006). The circulation due to combined Rossby and Kelvin wave responses is responsible for the El Niño and La Niña events known as the El Niño Southern Oscillation (ENSO) (Trenberth, 1997; Nishimoto and Shiotani, 2012; Scherllin-Pirscher et al., 2012). The intra-seasonal variation in the tropics, known as the Madden–Julian Oscillation (MJO), tends to be more active toward the Eastern Pacific during El Niño events (Son et al., 2017). The static stability in the upper troposphere tends to decrease with the deep convection associated with the MJO (Nishimoto and Yoden, 2017).

The vertical profile of $N^2$ across the tropopause (i.e., the sharpness) and the thickness of the layer of maximum $N^2$ above the tropopause have been determined in previous studies using both ground- and satellite-based observations. For example, Bell and Geller (2008) analyzed the twice daily standard radiosonde data from WMO stations and found that the thickness was ~1 km at low latitudes. Using data from the CHAllenging Mini satellite Payload (CHAMP) Global Navigation Satellite System Radio Occultation (GNSS-RO) mission (Wickert et al., 2001), Schmidt et al. (2005) reported that the tropopause sharpness varies less throughout the year in the tropics than in polar regions. Ratnam et al. (2005), also using the CHAMP dataset, reported an association between greater sharpness and a higher and colder tropopause.

Since the launch of Constellation Observing System for Meteorology Ionosphere and Climate (COSMIC) GNSS-RO in April 2006 (Anthes, 2011), which has a much greater number of occultations than CHAMP, investigation of the global characteristics of the tropical tropopause has intensified. The tropopause sharpness is defined as the difference between the mean $N^2$ ($\overline{N^2}$) in the region up to 1 km above the LRT or CPT ($\overline{N^2}_{+1}$), and $\overline{N^2}$ in the region down to 1 km below the LRT or CPT ($\overline{N^2}_{-1}$) (Kim and Son, 2012; Son et al., 2011). Son et al. (2011) showed maximum sharpness over the Western Pacific region in all seasons with slightly higher values during Northern Hemisphere (NH) winter, while Kim and Son (2012)





reported that the local structure and seasonal variability of the tropopause sharpness are associated with convectively coupled equatorial waves. Averaging the $N^2$ profile within ±1 km of the LRT or CPT can reduce fluctuations caused by small-scale perturbations. Kedzierski et al. (2016) described the role of Kelvin, inertia–gravity, and Rossby waves in modulating the maximum $N^2$ ($maxN^2$) above the LRT in the tropics using the long-term COSMIC dataset.

Fine-scale temperature profiles ($T$) from GNSS-RO measurements commonly have ~1 km vertical resolution in the upper troposphere and lower stratosphere (UTLS) (Kursinski et al., 1997). Support from the COSMIC Data Analysis and Archive Centre (CDAAC) allowed us to retrieve the dry temperature from COSMIC with higher resolution, reaching 0.1 km, in the UTLS. We are motivated to utilize the long-term COSMIC GNSS-RO data at a vertical resolution of 0.1 km to investigate

the annual, interannual, and intra-seasonal variability in properties of the tropical tropopause; i.e., the static stability and TIL parameters (sharpness and thickness). We begin by describing the latitudinal and longitudinal distributions of $N^2$ with respect to height, and both relative to the LRT height and CPT height. After displaying $N^2$ distributions in the three height coordinates, we investigate the relationships between TIL and ENSO variability, and between TIL and MJO activity.

**2 Data**

We briefly explain the difference between dry temperature from two COSMIC datasets, ***cosmicfsi*** and ***cosmic2013***, and their comparison with radiosonde data. We use ***cosmic2013*** only from October 2011 to March 2012 (183 days), the duration of the international collaborative campaign called the cooperative Indian Ocean experiment on intra-seasonal variability in the year 2011 and the joint project of Dynamics of the Madden–Julian Oscillation (CINDY-DYNAMO 2011) (Yoneyama et al.,

2013; Zhang et al., 2013). For the long-term analysis, we utilize ***cosmicfsi*** from 2007 to 2016 (10 years) to derive $N^2$ and determine the TIL parameters relative to the CPT. In the following, we outline details of high-resolution $T$ profiles, TIL definitions, and supporting information.

**2.1 High-resolution temperature profiles of COSMIC GNSS-RO**

The GNSS limb soundings received by low Earth orbit satellites that pass through the lower stratosphere down to the troposphere may contain multipath atmospheric signals, such as atmospheric minor constituents and sharp gradients at the tropopause (Melbourne, 2004). COSMIC GNSS-RO provides 1500–2000 profiles per day (~400 profiles over 10°S–10°N) (Anthes, 2011). The COSMIC retrievals apply wave optics algorithms such as Full Spectrum Inversion (FSI) and Phase Matching (PM) (Jensen et al., 2003, 2004), transforming the entire phase and amplitude of the occultation signal to derive

the dry atmospheric temperature profiles preserving vertical resolution of ~0.1 km at lower altitudes (Anthes, 2011; Gorbunov, 2002).

The ***cosmicfsi*** datasets use FSI up to 30 km altitude, but reliable $T$ profiles representing small perturbations in the lower stratosphere due to atmospheric gravity waves are limited to below ~28 km due to increasing noise from the retrieval data

processing (Tsuda et al., 2011). At altitudes above 30 km, $T$ of ***cosmicfsi*** is obtained from the time derivative of the excess phase of the GNSS signal (the geometrical optic method; Kursinski et al., 1997), which was used above 20 km for COSMIC data re-processed by CDAAC (***cosmic2013***). The ***cosmic2013*** dataset was retrieved using PM and smoothed over a 0.5 km scale in the range 10–20 km (Sokolovskiy et al., 2014; Zeng et al., 2016). There was good agreement between the $T$ profiles in the UTLS for both COSMIC datasets and radiosonde data, and the mean difference of CPT temperatures was within ±0.4

K (Noersomadi and Tsuda, 2017). The ***cosmicfsi*** dataset is freely available on the Inter-university Upper atmosphere Global



Observation NETwork (IUGONET) system of the metadata database of the Japanese inter-university research program (Hayashi et al., 2013).

We found 2,312 and 3,387 occultation profiles (~12 profiles/day and ~18 profiles/day) in ***cosmicfsi*** and ***cosmic2013***,
respectively, inside 90–150°E, 10°S–10°N (the Maritime Continent region) within a 183-day period. Note that the difference in the total number of occultations in the two retrievals is possibly caused by different truncations of the GNSS signal in the lower atmosphere (Schreiner et al., 2011), as reported by Noersomadi and Tsuda (2017). To compare the shape of $N^2$ profiles derived from $T$ by ***cosmicfsi*** and ***cosmic2013*** that have different vertical resolution, we use radiosonde observations collected during the CINDY-DYNAMO 2011 campaign. Table 1 lists the 13 stations in the Maritime Continent where
approximately twice-daily routine balloon soundings with 2-s recording were conducted. The total number of radiosonde balloons lifted successfully up to >2 km above the CPT is 3,996 (~21 profiles/day). All radiosonde data were downloaded from the Earth Observing Laboratory (http://data.eol.ucar.edu).

We adjusted $T$ from GNSS-RO in the geometrical height domain to the geopotential height used for radiosonde data before
performing the comparison. Figure 1 shows a typical COSMIC profile on 24 November 2011 at 12:43 UT, which is located within ~115 km horizontal radius from the balloon observation launched at 11:35 UT at Surabaya station (112.75°E, 7.31°S). Assuming the average ascent rate of the balloon is ~5 m s$^{-1}$ (Gong and Geller, 2010), the temperature measurement at 18 km occurred at around 12:35 UT, so the actual time difference with the occultation event is less than 30 min. The $T$ from ***cosmicfsi*** agrees very well with the radiosonde result, particularly above 16.5 km. The ***cosmicfsi*** shows small-scale
perturbations and large $N^2$ above the CPT, as shown in the radiosonde data.

The mean $N^2$ profiles of all available ***cosmicfsi***, ***cosmic2013***, and radiosondes within a 183-day period are shown in Fig. 2 (left). The $N^2$ profile just above the CPT from ***cosmicfsi*** that agreed very well with the radiosonde data is sharper than the $N^2$ from ***cosmic2013***. Figure 2 (right) shows the zonal mean of $N^2$ over 10°S–10°N latitude throughout the 10 years of ***cosmicfsi***
data (329,396 profiles). The existence of a sharp thin layer is seen from the long-term ***cosmicfsi*** dataset. This study mainly discusses the sharpness ($S$-ab, where 'ab' means 'above' and 'below' the CPT) and thickness ($dH$), which are defined in Fig. 2 (right panel).

### 2.2 TIL definitions

There are several definitions of TIL related to $N^2$ behavior. Schmidt et al. (2005) and Son et al. (2011) defined the TIL with respect to LRT height (more recently, Gettelman and Wang, 2015; Kedzierski et al., 2016). Ratnam et al. (2005) and Kim and Son (2012) defined the tropopause sharpness as the change in temperature gradient across the CPT. Considering that CPT is the appropriate reference for the tropical tropopause (Kim and Alexander, 2015), we define TIL parameters relative to CPT height in this study.
We obtain TIL parameters from the individual $N^2$ profiles, limiting the height range to ±3 km relative to the CPT, where most of the maximum and minimum peaks of $N^2$ are located. Definitions of the TIL parameters are summarized in Table 2. We define TIL sharpness as follows:

   1) $S$-aCPT is the difference between $maxN^2_{+1}$ and $N^2$ at the CPT,
2) $S$-bCPT represents the difference between $N^2$ at the CPT and minimum $N^2_{-1}$ ($minN^2_{-1}$), and
   3) $S$-ab the difference between $maxN^2_{+1}$ and $minN^2_{-1}$ (i.e., $S$-ab is equal to the sum of $S$-aCPT and $S$-bCPT).

The TIL thickness definitions include:




1) $dH$-aCPT and $dH$-bCPT are the corresponding distances of $maxN^2_{+1}$ and $minN^2_{-1}$ relative to the CPT, respectively,

2) $dH$-ab is the distance between height of $minN^2_{-1}$ and height of $maxN^2_{+1}$, and

3) $dH$ is the thickness over which $N^2 \geq 80\% maxN^2$

The TIL parameters are sorted on a $5° \times 5°$ longitude and latitude grid. The grid cells with no available TIL data are denoted as missing values. We analyze the frequency distribution and the climatology of TIL parameters in the region 0°–360°E, 30°S−30°N in the next section. To investigate the interannual and intra-seasonal variations, we focus on the low latitudes, 10°S−10°N. Due to the limited number of COSMIC profiles in each grid cell (1 or 2), we then averaged 5 days for data for
each grid cell recursively to determine the daily series data for investigating the TIL variations at intra-seasonal time-scales.

**2.3 Additional data**

To complement the analysis of TIL variations in the tropics, we use NOAA Outgoing Long Wave (OLR) data, which are considered a proxy for convective activity (Liebmann and Smith, 1996). We refer to the ENSO sea surface temperature
(SST) Niño 3.4 index (accessed from http://www.cpc.ncep.noaa.gov/data/indices/ on 9 August 2018) to investigate the influence of El Niño and La Niña on convective activity and TIL variability at an interannual time-scale. When analyzing the intra-seasonal variations, we use the OLR MJO Index (OMI) (Kiladis et al., 2014; available at https://www.esrl.noaa.gov/psd/mjo/mjoindex/), which consists of the first two principal components ($PC_1$ and $PC_2$) of the empirical orthogonal functions of 30–96 day filtered OLR. OMI is considered as the index that represents the eastward
propagation of deep convection (Kiladis et al., 2014; Nishimoto and Yoden, 2017). An MJO active phase is defined when the amplitude of OMI ($\sqrt{PC_1+PC_2}$) $\geq 1$; otherwise, MJO is inactive.

**3 Results**

**3.1 Global distribution of $N^2$**

Figure 3 shows the mean $N^2$ in conventional height coordinates at 5−25 km altitude as a function of latitude from 2007 to 2016 in two longitude regions with different land−sea distribution: the Maritime Continent (MC; 90°−150°E) and the Pacific Ocean (PO; 170°−230°E) during the NH winter (December–January–February; DJF) and NH summer (June–July–August; JJA). In all panels, we also display the mean LRT and CPT heights. The discontinuity of the mean LRT height from the tropics to extra tropics is obvious in DJF, especially in the MC region (Fig. 3a). The mean CPT heights in the subtropics,
around 20°–30°N and 20°–30°S in the MC, are 0.2–0.4 km higher than at the equator (Fig. 3a and c). However, in the PO the mean CPT height in the subtropical winter hemisphere is somewhat (~0.2 km) higher than at the equator and in the subtropical summer hemisphere. The results shown here are much smoother than those reported by Ratnam et al. (2005) using CHAMP data. Figure 3a–d indicates that the mean LRT height (~16.5 km) is ~0.7 km lower than the mean CPT height (~17.2 km) over 20°S–20°N.

The LRT heights over the PO and the Atlantic and Indian oceans are 0.2–0.4 km lower than those over the MC (Fig. 4). The lower LRT over the ocean regions may be due to the definition of LRT, which depends on the vertical temperature gradient (d$T$/d$z$), which is sensitive to the height resolution (Noersomadi and Tsuda, 2017). When increasing temperatures appear below 15 km, as found by Fujiwara et al (2003) who used radiosonde data in the Eastern Pacific, d$T$/d$z$ may increase
significantly so that the mean d$T$/d$z$ within 2 km above that level satisfies the WMO definition. On the other hand, when



there are no significant increasing temperatures below 15 km, the LRT should be the same as the lowest CPT, around 16–17 km.

The $N^2$ above the tropical tropopause in DJF is larger than in JJA in Fig. 3. This is also seen in the height–longitude cross-
section over 10°S–10°N in Fig. 4. The mean $N^2$ has a maximum of about $8 \times 10^{-4}$ s$^{-2}$ at 17.5−19.0 km height. This height range is located around 1–2 km above the mean LRT height. The $maxN^2$ of individual profiles are located mainly within 1 km above the CPT height. Averaging $N^2$ in conventional height coordinates tends to smooth out the tropopause sharpness (Birner et al., 2002). Therefore, we describe the mean $N^2$ relative to both the LRT and CPT heights in each profile.

Figure 5 shows a height–latitude cross-section of the mean $N^2$ at ±3 km relative to the LRT height (the $y$-axis is the LRT-relative coordinate). The maximum $N^2$ around 1–2 km above the LRT in the polar summer hemisphere has been observed using CHAMP data (Randel et al., 2007; Schmidt et al., 2010). Randel et al. (2007) reported that the large-scale background temperature above the LRT may influence the seasonal cycle of the TIL in the polar region. The $N^2$ values above the LRT in the 20°S–20°N latitude range for DJF and JJA in the two longitude regions are $\geq 6.0 \times 10^{-4}$ s$^{-2}$ and generally consistent with
earlier results (Grise et al., 2010).

In contrast to previous studies, our results show a thin layer (about 0.3–0.5 km thick) just above the LRT in both seasons and longitude regions with $N^2 \geq 4.0 \times 10^{-4}$ s$^{-2}$ over the entire latitude range. The thin layer with $N^2 \geq 7.0 \times 10^{-4}$ s$^{-2}$ in the tropics over the MC covers a broader area than over the PO, particularly in DJF (Fig. 5a and b). The region with $N^2 \geq 7.0 \times 10^{-4}$ s$^{-2}$
during NH winter around 10°–20°N and 10°–20°S expands up to 3 km above the LRT in the MC, while it is limited near 2 km above the LRT in the PO. We observe a twin enhancement of large $N^2$ in the tropics, the lower one within 1 km of the LRT and the second one ~2 km above the LRT. The results also show a clearer $maxN^2$ in the Southern Hemisphere (SH) winter, limited around 40°–50°S. Values of $N^2$ in the range $3.5$–$4.5 \times 10^{-4}$ s$^{-2}$ in the extratropics above the LRT separate the regions of large $N^2$ in the tropics and polar regions.

In Fig. 6, we investigate the mean $N^2$ in the coordinate relative to CPT height from the same data by focusing on the 30°S–30°N latitude range. The twin enhancement of $N^2$ above LRT is not seen after changing the relative height coordinate. During NH winter and NH summer, both longitude regions have a single layer of ~0.5 km thick with large $N^2$ of ~$12.0 \times 10^{-4}$ s$^{-2}$. The area of $N^2 \geq 10.0 \times 10^{-4}$ s$^{-2}$ in DJF extends over 15°S–15°N over the MC (Fig. 6a), while it is limited
to around 10°S–10°N over the PO (Fig. 6b). The area with $N^2 \geq 10.0 \times 10^{-4}$ s$^{-2}$ in JJA is generally narrower than that in DJF. During DJF over the MC, values of $N^2$ of about $7.0$–$8.0 \times 10^{-4}$ s$^{-2}$ are found over 10°–20°N and 10°–20°S between 1 and 2 km above the CPT. A similar pattern is seen over the PO but the enhancement of $N^2$ around 20°N and 20°S is smaller than that in the MC. By changing the coordinate from relative to LRT height, to relative to CPT height, we find that $maxN^2$ changes from $8.0 \times 10^{-4}$ s$^{-2}$ to $12.0 \times 10^{-4}$ s$^{-2}$.

The profiles of large $N^2$ over 20°N and 20°S in the MC region represent the vertical section of the Kelvin and mixed Rossby–gravity waves response known as the Matsuno–Gill pattern mode (Matsuno, 1966; Gill, 1980; Grise et al., 2010; Nishimoto and Shiotani, 2012). The vertical propagation of equatorial waves (Kelvin waves and/or gravity waves) modulates the tropopause (Tsuda et al., 1994; Kim and Alexander, 2015). When there is a single layer of minimum temperature, the
LRT and CPT are at the same location/height. On the other hand, when the coldest $T$ is located at the highest of multiple temperature minima, the lowest minimum is determined as the LRT (Noersomadi and Tsuda, 2017). Therefore, between the LRT and CPT, $N^2$ tends to decrease because $T$ is decreasing toward its coldest value. The combination of large $N^2$ above the





LRT (single cold peak) and low $N^2$ above the LRT (multiple cold peaks) will tend to smear out the average of $maxN^2$ above the LRT.

We show height–longitude cross-sections of the mean $N^2$ at 10°S–10°N in NH winter and NH summer relative to LRT

height and CPT height in Fig. 7 and Fig. 8, respectively. Large $N^2$ is found in DJF and JJA above the LRT (Fig. 7). The $N^2$ decreases to $4.5 \times 10^{-4}$ s$^{-2}$ around 0.5–1.0 km above the LRT after increasing in the first layer of large $N^2$ (0.0–0.5 km above the LRT). The decrease in $N^2$ at this height range is clearly seen in the oceanic regions 30°–90°E, 180°–240°E and 300°– 350°E (Fig. 7a). The enhancement of $N^2$ at 2 km above the LRT differs significantly between DJF and JJA, as shown by contour lines $N^2 \geq 7.0 \times 10^{-4}$ s$^{-2}$. In DJF, the layer of large $N^2$ located around the MC and PO is larger and thicker than the

$N^2$ near Africa and South America. These results show the different characteristics of the inversion layer above the LRT over land and ocean regions.

Figures 5 and 7 show a decrease of $N^2$ at 0.5–1.0 km above the LRT, between the two regions of elevated $N^2$, in particular over the oceanic regions. These are probably related to increasing temperature in the upper atmosphere (Fujiwara et al.,

2003; Nishi et al., 2010). The level of increasing temperature can be determined as the LRT height when the WMO definition is attained. Because the temperature decreases above the LRT to reach its coldest point, $N^2$ also decreases. Hence, the $N^2$ values within 0.5–1.0 km are smoothed. When the CPT-relative height coordinate is used, the region of elevated $N^2$ becomes a single thin layer with maximum value reaching $12.0 \times 10^{-4}$ s$^{-2}$ (Fig. 6). Values of $N^2 \geq 10.0 \times 10^{-4}$ s$^{-2}$ are more prominent in DJF than in JJA. Values of $N^2 \geq 7.0 \times 10^{-4}$ s$^{-2}$ at 1–2 km above CPT are shifted somewhat eastward from the

maximum at 0.0–0.5 km above the CPT, especially at 120°–210°E and 240°–300°E (Fig. 8).

Figures 6 and 8 show a thin layer of low $N^2$ within the 1 km layer below the CPT and large $N^2$ within the 1 km above the CPT. In the following section we investigate the frequency distribution of TIL parameters related to the shallow layer within ±1 km of the CPT.

### 3.2 Frequency Distribution of TIL Parameters

The histograms of all TIL parameters over the period 2007–2016 for the entire longitude range at 30°S–30°N are presented in Fig. 9. All parameters show a skewed distribution. In Fig. 9 (left column), the mean $S$-aCPT is larger than $S$-bCPT. The values of $S$-aCPT smaller than its mean are mostly in the range $2.8-6.6 \times 10^{-4}$ s$^{-2}$, and the values larger than its mean in the

range $8.4-18.0 \times 10^{-4}$ s$^{-2}$. About 80% of $S$-bCPT values lie in the range $1.7-6.2 \times 10^{-4}$ s$^{-2}$, while only about 3% have $S$-bCPT $> 8.0 \times 10^{-4}$ s$^{-2}$. The mean and standard deviation of $S$-ab are $(10.5 \pm 3.7) \times 10^{-4}$ s$^{-2}$. The tail to the right of the mean $S$-ab is longer than that to the left (i.e., positive skewness). The results indicate a large variation in temperature gradient within 1 km above the CPT. The natural question would be when and where large values (the longer tails) appear and what atmospheric processes are responsible. In the next section, we discuss the variation in $S$-ab that is associated with convective

activity.

The distribution of $dH$-aCPT, which has mean and standard deviation $0.4 \pm 0.2$ km, is similar to that of $S$-aCPT, while $dH$- bCPT is nearly symmetric with mean and standard deviation $0.5 \pm 0.2$ km (Fig. 9 right column). About 60% of the values of $dH$-aCPT lie in the range 0.2–0.4 km. That means that 60% of $maxN^2$ are located between the CPT and 0.4 km above it.

About 70% of the magnitudes of $dH$-ab are within $1.0 \pm 0.2$ km. This is reasonable because $dH$-ab is equal to the sum of $dH$- aCPT and $dH$-bCPT, and this means the distance between the $minN^2$ level below the CPT and the $maxN^2$ level above CPT was 1 km. The $minN^2$ and $maxN^2$ are related to decreasing and increasing temperature, respectively. Decreasing and





increasing temperature near the CPT could be influenced by Kelvin wave activity that might affect the stratospheric and tropospheric air exchange (Tsuda et al., 1994; Fujiwara and Takahashi, 2001).

The thickness $dH$ varies between 0.3 and 0.6 km, but 70% of the values are within $0.4 \pm 0.04$ km (green histogram in Fig. 9,

5  lower right panel). This result shows a very thin TIL layer above the CPT in the tropics. The thin layer of TIL can also be seen in Fig. 2 (right) and Fig. 6. If the mean $maxN^2$ is in the range $11–12 \times 10^{-4}$ s$^{-2}$ in both seasons, then 80% of the $maxN^2$ is about $8.8–9.6 \times 10^{-4}$ s$^{-2}$. The contour lines of $N^2$ at this interval depict a very thin layer (<0.5 km) in both longitude regions (Fig. 6). This leads to the question of how the thickness varies in the tropics. In the following section, we investigate the climatology of $dH$.

### 3.3 Seasonal Variations in TIL Sharpness and Thickness

Figure 10 shows horizontal distributions of the mean $S$-ab, $dH$, and OLR in the tropics during DJF and JJA. The mean $S$-ab in the low latitudes during DJF is larger than that during JJA. The highest values, up to $16–18 \times 10^{-4}$ s$^{-2}$, are associated with low OLR values, suggesting strong deep convection over the following convective regions: (i) Africa, (ii) a wide area from

the Maritime Continent to the Western Pacific, and (iii) South America. Large $S$–ab around the 240°–270°E longitude region in DJF is qualitatively related to OLR values of 220–240 W m$^{-2}$ representing the inter tropical convergence zone (ITCZ). Large $S$-ab values of $14–16 \times 10^{-4}$ s$^{-2}$ are also found over South Asia and near the ITCZ in JJA. Radiative effects from cirrus cloud may be responsible for the enhancement of $S$-ab. Sassen et al. (2009) demonstrated that cirrus clouds are confined to the monsoon region and ITCZ where they are generated in anvils created by deep convection. The cirrus cloud decreases the

static stability below the tropopause (Nishimoto and Yoden, 2017; Son et al., 2017). Therefore, decreasing static stability below the tropopause will tend to enlarge the difference between $maxN_{+1}$ and $minN^2_{–1}$.

Bell and Geller (2008) investigated the thickness defined as the distance from the LRT to the level where $dN^2/dz$ reached its minimum in the stratosphere ($z$ is the vertical coordinate) using global interpolated radiosonde data. They reported a

thickness of ~1 km at latitudes around 15°N. Since there are few radiosonde stations in the low latitudes, they did not show the horizontal distribution of the thickness. Here, we use a different definition of $dH$ as explained in Section 2. In Fig. 10 (middle row), values of $dH$ lie in the range 0.39–0.48 km. Large $dH$ associated with low OLR values over the three convective regions agrees qualitatively with large $S$-ab.

Deep convection over Indonesia and northern Australia near Darwin during the Australian monsoon (DJF) and over the Bay of Bengal, South Asia, and the Philippines during the Asian monsoon (JJA) seems to be the main control on $S$-ab and $dH$. To emphasize the effect of convective activity, we show the mean $N^2$ profiles in DJF and JJA in Fig. 11. We define 90°–140°E, 10°S–0 in DJF and 80°–130°E, 10°N–20°N in JJA as the "convective" regions (OLR ≤ 240 W m$^{-2}$). The "non-convective" regions (OLR > 240 W m$^{-2}$) are 200°–250°E, 10°S–0 in DJF and 190°–240°E, 15°N–25°N in JJA. We selected those regions

as described in Fig. 10 (bottom row), each with the same area (50° longitude × 10° latitude). The total numbers of profiles from 10 years of COSMIC GNSS-RO observation inside the convective regions are 9,140 in DJF and 18,046 in JJA, while inside the non-convective regions there are 8,758 profiles in DJF and 17,103 in JJA. COSMIC satellites operate in polar orbits, so the highest number of profiles are located in mid latitudes (e.g., fig. 1 in Son et al., 2011). The total numbers of profiles in JJA are larger than in DJF because the convective and non-convective regions in JJA are defined to lie closer to

the mid-latitudes. It is clear that in both DJF and JJA the mean $N^2$ over the convective regions is smaller below the CPT and larger above CPT than the mean $N^2$ over the non-convective regions. The difference of the mean $N^2$ is $\sim 0.5 \times 10^{-4}$ s$^{-2}$ below the CPT and $\sim 1.3 \times 10^{-4}$ s$^{-2}$ near the level of $maxN^2$ above the CPT. The observations show that convective activity resulted



in decreased $N^2$ below CPT and increased $maxN^2_{+1}$, which increased $S$-ab from $9.4 \times 10^{-4}$ s$^{-2}$ to $10.4 \times 10^{-4}$ s$^{-2}$ and tended to increase $dH$ from 0.3 to 0.5 km.

We have examined the spatial distributions of $S$-ab and $dH$ in boreal winter and summer. To show the seasonal cycle in the different longitude regions, we calculated the monthly mean and standard deviation of $S$-ab and $dH$ over 10°S–10°N as shown in Figs. 12 and 13. The sharpness and thickness are greater in MC than in PO. The sharpness in MC during July–August is somewhat higher than in May–June and September–October, while it is relatively constant at $11 \times 10^{-4}$ s$^{-2}$ in PO from March to November. It is also clear that the seasonal cycle in MC is affected by sub-seasonal variation. On the other

hand, the thickness in PO shows a clear annual cycle with a minimum in JJA. The results indicate that land–sea distribution influences the variability in $S$-ab and $dH$.

In the following section we discuss the interannual and intra-seasonal variability of $S$-ab.

**3.4 Influence of El Niño and La Niña**

We investigate the monthly mean $S$-ab as well as OLR data in the two longitude regions (Fig. 14). Large $S$-ab in the MC associated with low OLR values can be seen during NH winter as part of the annual cycle. Neither the annual peak of $S$-ab nor the peak of low OLR over the MC is seen in DJF for 2008–2009. On the other hand, $S$-ab increased from ~$11 \times 10^{-4}$ s$^{-2}$ to $12 \times 10^{-4}$ s$^{-2}$ over the PO in the same period. This suggests a signal of interannual variation in $S$-ab and OLR in both the

MC and PO. Another signal that can be interpreted as interannual variation is seen in DJF of 2014–2015, where $S$-ab showed a strong peak and convective activity was also enhanced in the PO. The linear correlation between $S$-ab and OLR in the MC and PO has coefficients of −0.57 and −0.67, respectively, delineating an out-of-phase relation.

The negative correlation from the monthly mean time series may still be affected by the annual cycle or intra-seasonal

variation. Therefore, we investigate the $S$-ab and OLR anomalies ($S$-ab* and OLR*) calculated by subtracting the climatological values from the monthly mean time series. Next, we compare both $S$-ab* and OLR* with the SST Nino 3.4 index (Fig. 15). We applied a 4-month running mean to smooth the time series and reduce the possible intra-seasonal fluctuation, as subtracting the climatological mean will only remove the annual cycle. El Niño events were defined when SST Niño 3.4 > +0.5. Strong El Niño events occurred during the NH winters 2008–2009 and 2014–2015. La Niña events

were determined when SST Niño 3.4 < −0.5. La Niña events were seen during the NH winters 2006–2007, 2007–2008 and 2009–2010. A linear correlation analysis shows that $S$-ab* is negatively correlated with both OLR* (−0.56) and SST Niño 3.4 (−0.66) in the MC. Surprisingly, in the PO, $S$-ab* is strongly negatively correlated with OLR* (−0.90) and strongly positively correlated with SST Niño 3.4 (+0.88). The negative and positive correlations between $S$-ab* and SST Nino3.4 show that El Niño and La Niña phenomena have a significant impact on the variation in $S$-ab*. The negative correlation

between $S$-ab* and OLR*, which is modulated by SST Niño 3.4, means that convective activity on the interannual time-scale influences $S$-ab* in both longitude regions.

To analyze in more detail the influence of El Niño and La Niña, we calculate the mean $N^2$ profile over the MC and PO during DJF El Niño and DJF La Niña (Fig. 16). The mean $N^2$ during La Niña events below the CPT was smaller and above

the CPT was larger than during El Niño events over the MC, as more convection occurred. Under La Niña conditions, the mean $maxN^2$ overshoots up to $12 \times 10^{-4}$ s$^{-2}$ and relaxes gradually to the lowermost stratospheric value of $6 \times 10^{-4}$ s$^{-2}$ at 2 km above the CPT. However, under El Niño conditions the mean $maxN^2$ is ~$10 \times 10^{-4}$ s$^{-2}$ and sharply decreases to $6 \times 10^{-4}$ s$^{-2}$



about 0.5 km above the CPT. Convective clouds that are more active during El Niño in the PO than during La Niña are the cause of the difference in mean $N^2$ profiles between these two cases shown in Fig. 16 (middle). The $maxN^2$ in El Niño was $11.4 \times 10^{-4}$ s$^{-2}$, larger than in La Niña ($10 \times 10^{-4}$ s$^{-2}$), but in both cases the values of $N^2$ sharply decrease then coincide at the lowermost stratospheric value at ~1 km above the CPT. The results show that convective activity in both longitude regions

during El Niño and La Niña will extend the $maxN^2_{+1}$ above the CPT so that $S$-ab increases from $9.4 \times 10^{-4}$ s$^{-2}$ to $10.9 \times 10^{-4}$ s$^{-2}$.

Figure 16 (right) shows the difference in the mean $N^2$ ($\Delta N^2$) between El Niño and La Niña events. In the MC and PO, the $\Delta N^2$ below the CPT is ~$0.5 \times 10^{-4}$ s$^{-2}$ in both cases. The surprising result is that above the CPT the peak amplitude $\Delta N^2$ in the

MC is higher and larger (0.5 km above the CPT and $1.8 \times 10^{-4}$ s$^{-2}$) than in the PO (0.2 km above CPT and $1.4 \times 10^{-4}$ s$^{-2}$). The difference in thermal structure over land and ocean may cause the difference in peak amplitude of $\Delta N^2$ above the CPT. The possible physical processes are as follows. Warmer SST under La Nina conditions results in more active convection that tends to force air upward to the TTL and then decrease the static stability below the CPT. The stratospheric air resists a change in height, so the temperature gradient becomes greater, enhancing $maxN^2$ above the CPT. Although our analysis is

based on limited observations, the results indicate that tropopause sharpness is correlated with convective activity associated with El-Niño and La-Niña phenomena.

### 3.5 MJO modulation

The convective activity in the tropics is also influenced by intra-seasonal variability (Wheeler and Kiladis, 1999). We extend

our analysis to the intra-seasonal variation in tropopause sharpness related to MJO propagation in the tropics. An MJO active phase is represented by large-scale deep convection that moves eastward, while an inactive phase is marked by suppressed convection (Zhang, 2005). Organized convective clouds propagate from the Indian Ocean to the Western Pacific from phase 3 to phase 6, as defined by Wheeler and Hendon (2004), over a period of ≤10 days or within 10−15 days during fast MJO episodes. During slow MJO episodes the propagation takes more than 20 days (Yadav and Straus, 2017). To identify fast or

slow MJO events, we use OMI data because they are derived directly from OLR and represent the convective signal of MJO (Kiladis et al., 2014). Figure 17 shows case studies for a slow MJO episode from 1 December 2007 to 28 February 2008 and a fast MJO episode from 15 January 2012 to 15 April 2012. The figure shows the propagation for both MJO active and inactive phases. In this figure, we obtained $S$-ab* by subtracting the average of $S$-ab over 120 days from the 5-day running mean time series. We applied a band-pass filter to the fluctuation components of $S$-ab with a cut-off frequency of 30–90 days

to retain MJO phase propagation (Kiladis et al., 2014; Zhang, 2005). The same analysis was applied to the OLR data.

Enhancement of $S$-ab* is obviously associated with eastward propagation of organized deep convection from the western Indian Ocean to the central Pacific during December 2007 and January 2008 (~60 days), which is classified as a slow MJO episode (Fig. 17 left). During the fast MJO episode (Fig. 17 right), eastward movement of large positive $S$-ab* also coincided

with negative OLR* from January 23 to February 8, 2012 (a period of 17 days). When the deep convection was suppressed during the inactive phase, $S$-ab* decreased. Hence, MJO propagation in the tropics has an impact on the variability in tropopause sharpness. A strong correlation between $S$-ab* and OLR* propagation on the intra-seasonal time-scale indicates that the organized deep convection reaches the upper troposphere and tends to enhance the temperature gradient in the TTL.

We investigated the $N^2$ profiles over the MC during the MJO active (5–15 January 2008) and MJO inactive (15–25 February 2008) phases. Figure 18 shows the characteristics of the difference in the mean $N^2$ between the periods of strong and weakened convection over the MC. The difference in the mean $N^2$ ($\Delta N^2$) between the MJO active and inactive phases shows



negative values of ~$1 \times 10^{-4}$ s$^{-2}$ at 1–3 km below the CPT, then jumps to $5 \times 10^{-4}$ s$^{-2}$ within 0.5 km above the CPT, suggesting that convection tends to lower the static stability below the CPT and increase the static stability just above it. This explains the large values of $S$-ab of $13.9 \times 10^{-4}$ s$^{-2}$ that form part of the long tail in the frequency distribution (Fig. 9).

**4 Concluding remarks**

COSMIC GNSS-RO data retrieved using Full Spectrum Inversion provide 0.1 km vertical resolution temperature profiles ($T$) in the UTLS. Using these data for 2007 to 2016, we investigated the global distribution of $N^2$ around the UTLS and the variation in the tropical TIL. We demonstrated the different characteristics of $N^2$ over the MC and PO. When the $N^2$ profiles are averaged relative to LRT height, we found twin peaks of $N^2$ in the tropics. The first is located within 1 km of the LRT,

and the second at ~2 km above the LRT. These twin peaks were not seen when the height relative to the CPT was used. The maximum of the mean $N^2$ changed from $8.0 \times 10^{-4}$ s$^{-2}$ to $12.0 \times 10^{-4}$ s$^{-2}$ within 1 km above the CPT. This was because vertical propagation of equatorial waves influences the temperature structure and thus affects the determination of the LRT and CPT. If the coordinate relative to the LRT is used, large and small values of $N^2$ above the LRT smeared out the average of $maxN^2$ above the LRT. Latitudinal and longitudinal distributions show that $N^2$ above the CPT in DJF was larger than in

JJA.

Since the CPT is the appropriate reference for the tropical tropopause, we analyzed the TIL parameters using $N^2$ profiles relative to the CPT. We found that ~60% of $maxN^2$ values were located within 0.5 km above the CPT. The frequency distribution of the tropopause sharpness ($S$-ab) has a positive skewness with a longer tail to the right of the mean than to the

left. The mean and standard deviation of $S$-ab are $(10.5 \pm 3.7) \times 10^{-4}$ s$^{-2}$. This suggests a large variation in the temperature gradient within $\pm 1$ km of the CPT. Seventy percent of thickness ($dH$) values are in the range $0.4 \pm 0.04$ km. Analysis of the seasonal variation shows that large $S$-ab and large $dH$ are associated with deep convection during the Australian and Asian monsoons. In convective regions the mean $N^2$ is smaller below the CPT and larger above the CPT than in non-convective regions.

We analyzed the interannual variation in $S$-ab ($S$-ab*) and OLR (OLR*) anomalies, and the SST Nino 3.4 index. We found an out-of-phase relation between $S$-ab* and OLR* over both the Maritime Continent (MC) and Pacific Ocean (PO), with correlation coefficients $-0.56$ and $-0.90$, respectively. $S$-ab* in the MC was negatively correlated with SST Niño 3.4 ($-0.66$), while in the PO it was in-phase and strongly correlated ($+0.88$). We also calculated the mean $N^2$ over the MC and

PO during DJF El Niño and DJF La Niña. The results indicate that during La Niña over the MC and El Niño over the PO, warmer SST gives more convection that tends to drive air upward in the TTL and increases the temperature gradient (positive $S$-ab*).

We also analyzed the intra-seasonal variation in $S$-ab*. Case studies during slow and fast MJO episodes show strong

eastward propagation of $S$-ab* associated with deep convection. Meanwhile, when organized deep convection is suppressed in the MJO inactive phase, $S$-ab* decreases. This suggests that the variations in $S$-ab* in the tropics are strongly related to the convective activity at this time-scale.

We have investigated the characteristics of the tropical TIL from the new definitions using the COSMIC GNSS-RO dataset

that has higher resolution in the UTLS. The tropical TIL may closely related to water vapor and ozone exchange between the



stratosphere and troposphere. Further discussion of the relationship between the TIL and radiative processes, as well as the dynamics of equatorial planetary-scale and small-scale gravity waves, is needed.

**Acknowledgements**

This work is supported in part by the Japan Society for the Promotion of Science (JSPS KAKENHI) Grant Number JP18K03741. Fruitful discussion with Prof. Shigeo Yoden is acknowledged. Author (N) received scholarship from Research and Innovation in Science and Technology Project (Riset-Pro), Ministry of Research, Technology and Higher Education (Kemenristekdikti) of Indonesia.

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





Table 1 Radiosonde stations for daily routine balloon launches during the CINDY-DYNAMO 2011 campaign around the Maritime Continent. All stations apart from No. 2 are in Indonesia. The 'Number of profiles' column lists the number of successful balloon soundings that reached >2 km above the CPT.

| No | Station name | Longitude | Latitude | Number of profiles |
|----|--------------|-----------|----------|--------------------|
| 1 | Singapore | 103.80°E | 1.30°N | 247 |
| 2 | Medan | 98.68°E | 3.57°N | 316 |
| 3 | Padang | 100.35°E | 0.88°S | 379 |
| 4 | Pangkal Pinang | 106.14°E | 2.16°S | 253 |
| 5 | Cengkareng | 106.68°E | 6.12°S | 303 |
| 6 | Ranai | 108.39°E | 3.91°N | 240 |
| 7 | Surabaya | 112.78°E | 7.37°S | 309 |
| 8 | Makassar | 119.53°E | 5.06°S | 307 |
| 9 | Palu | 119.91°E | 0.92°S | 337 |
| 10 | Manado | 124.92°E | 1.54°N | 334 |
| 11 | Ambon | 128.10°E | 3.71°S | 312 |
| 12 | Biak | 136.10°E | 1.19°S | 321 |
| 13 | Merauke | 140.41°E | 8.52°S | 338 |

Table 2 Definitions of TIL sharpness and thickness

| Parameter | Definition | Unit |
|-----------|------------|------|
| $S$-aCPT | $maxN^2_{+1} - N^2$ at CPT | $\times 10^{-4}$ s$^{-2}$ |
| $S$-bCPT | $N^2$ at CPT $- minN^2_{-1}$ | $\times 10^{-4}$ s$^{-2}$ |
| $S$-ab | $maxN^2_{+1} - minN^2_{-1}$ | $\times 10^{-4}$ s$^{-2}$ |
| $dH$-aCPT | $Hmax - H$ of CPT | km |
| $dH$-bCPT | $H$ of CPT $- Hmin$ | km |
| $dH$-ab | $Hmax - Hmin$ | km |
| $dH$ | $H$ of $80\% \times maxN^2$ above $Hmax - H$ of $80\% \times maxN^2$ below $Hmax$ | km |

$Hmax$: $H$ of $maxN^2_{+1}$; $Hmin$: $H$ of $maxN^2_{-1}$; subscript $\pm 1$ indicates within $\pm 1$ km





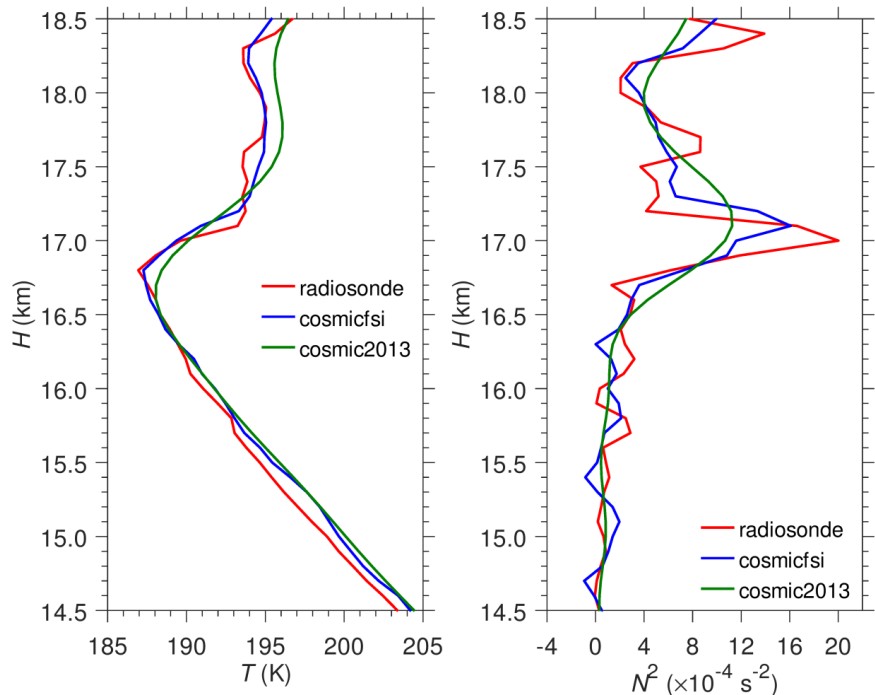

**Figure 1** Typical COSMIC GNSS-RO profile near the radiosonde balloon launch in Surabaya on 24 November 2011.



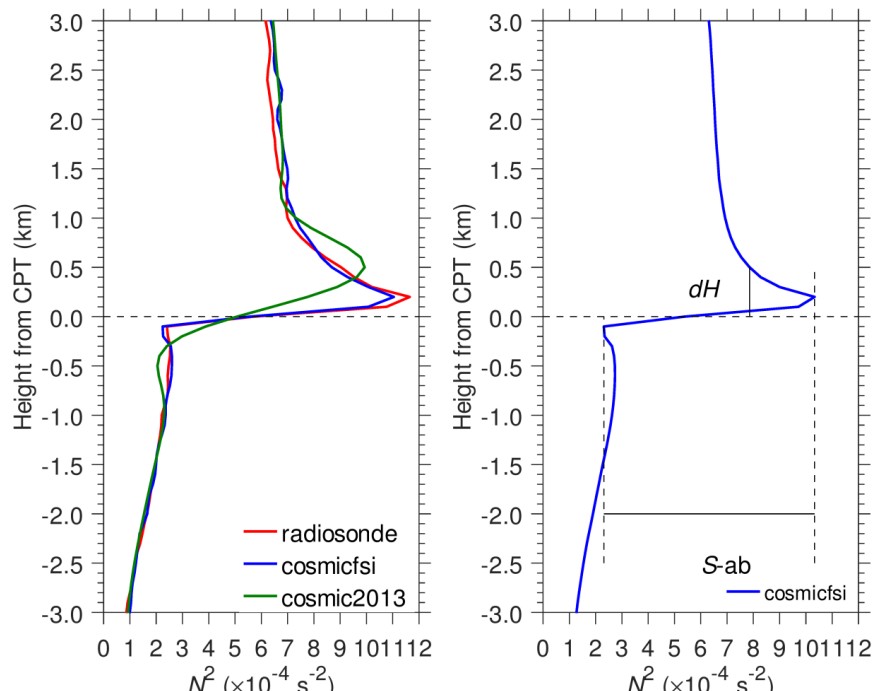

**Figure 2** Mean $N^2$ profiles from radiosonde (red), cosmicfsi (blue), and cosmic2013 (green) during CINDY-DYNAMO 2011 (left panel).
Zonal mean $N^2$ profile in the 10°S–10°N latitude range from 10 years cosmicfsi data (right panel). All heights are relative to the CPT.




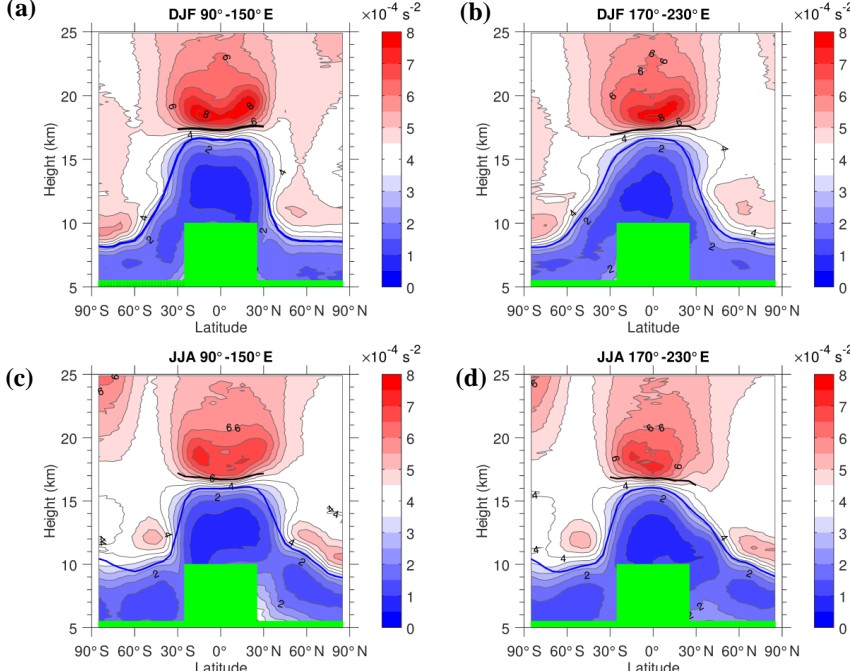

**Figure 3** Height–latitude distribution of the mean $N^2$ in conventional height coordinates for (a), (b) December–January–February (DJF) and (c), (d) June–July–August (JJA) in the two longitude regions. Color contour interval is $0.5 \times 10^{-4}$ s$^{-2}$. Thick black and blue lines indicate the mean CPT height and the mean LRT height, respectively. Green shading indicates where the dry temperature profiles of COSMIC show unrealistic values due to high humidity in the troposphere.





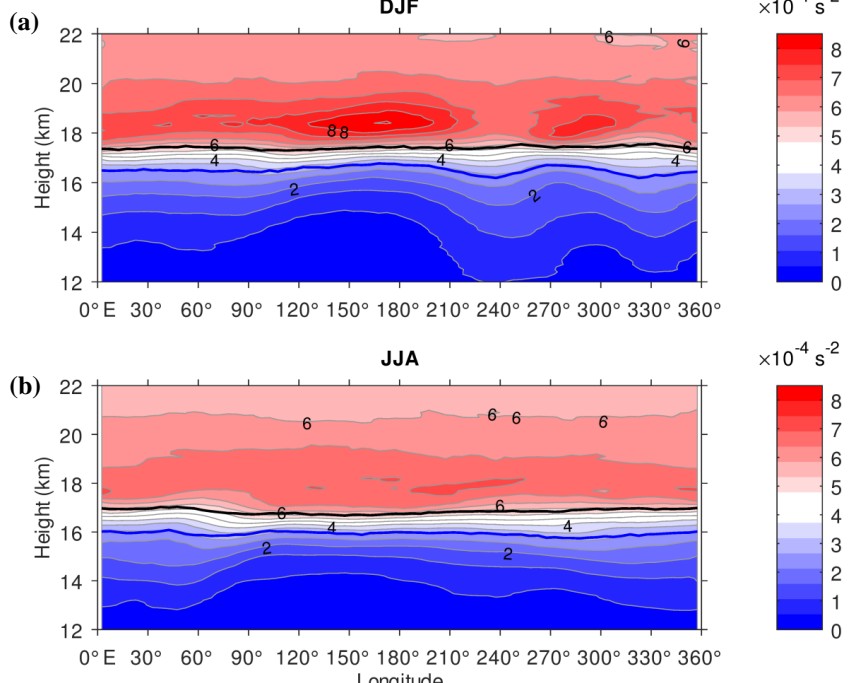

**Figure 4** Height–longitude cross-section of the mean $N^2$ over 10°S–10°N in conventional height coordinates for (a) DJF and (b) JJA.

Color contour interval is $0.5 \times 10^{-4}$ s$^{-2}$. Thick black and blue lines indicate the mean CPT height and the mean LRT height, respectively.





**Figure 5** Height–latitude distribution of the mean $N^2$ relative to the LRT location in the vertical for (a), (b) DJF and (c), (d) JJA in the two longitude regions. Color contour interval is $0.5 \times 10^{-4}\ \mathrm{s}^{-2}$.





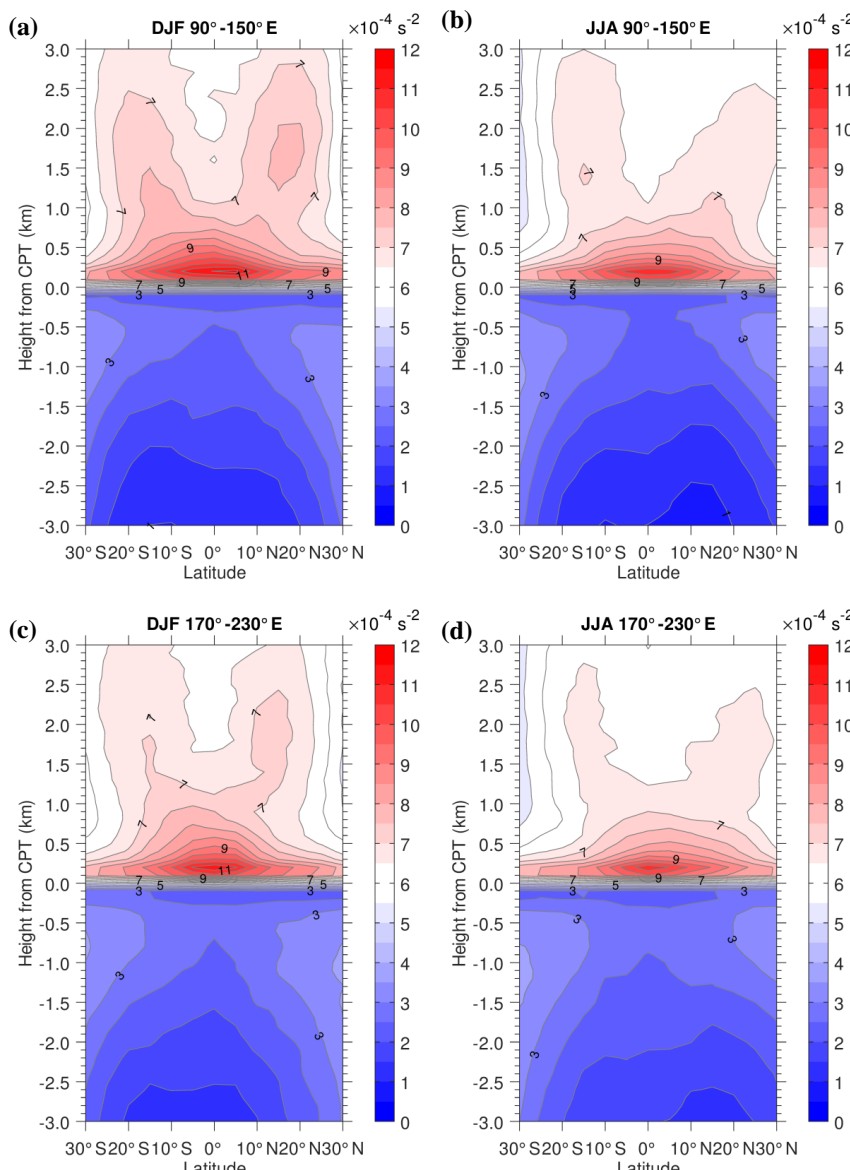

**Figure 6** Height–latitude cross-sections of the mean $N^2$ relative to the CPT location in the vertical for (a), (c) DJF and (b), (d) JJA in the two longitude regions. Color contour interval is $0.5 \times 10^{-4}\ s^{-2}$.



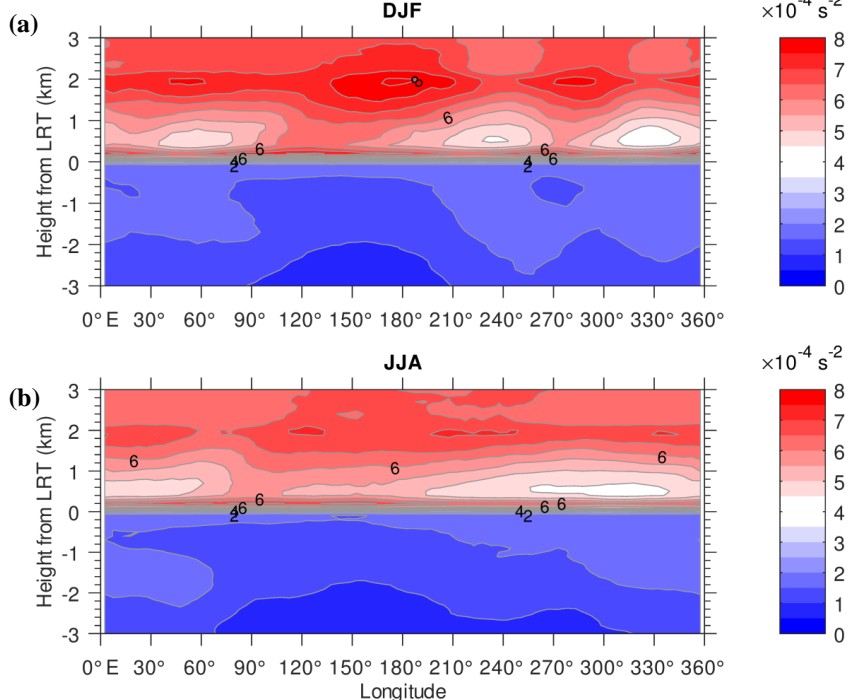

**Figure 7** Height–longitude cross-sections of the mean $N^2$ over 10°S–10°N latitude relative to the LRT location in the vertical for (a) DJF and (b) JJA. Color contour interval is $0.5 \times 10^{-4}\,\mathrm{s}^{-2}$.





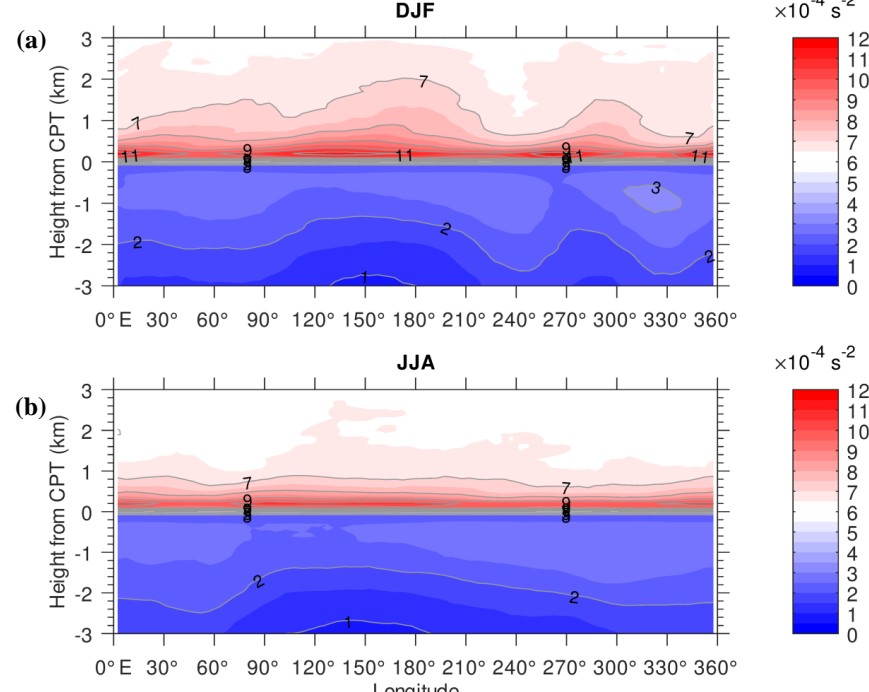

**Figure 8** As for Fig. 7, but relative to the CPT.





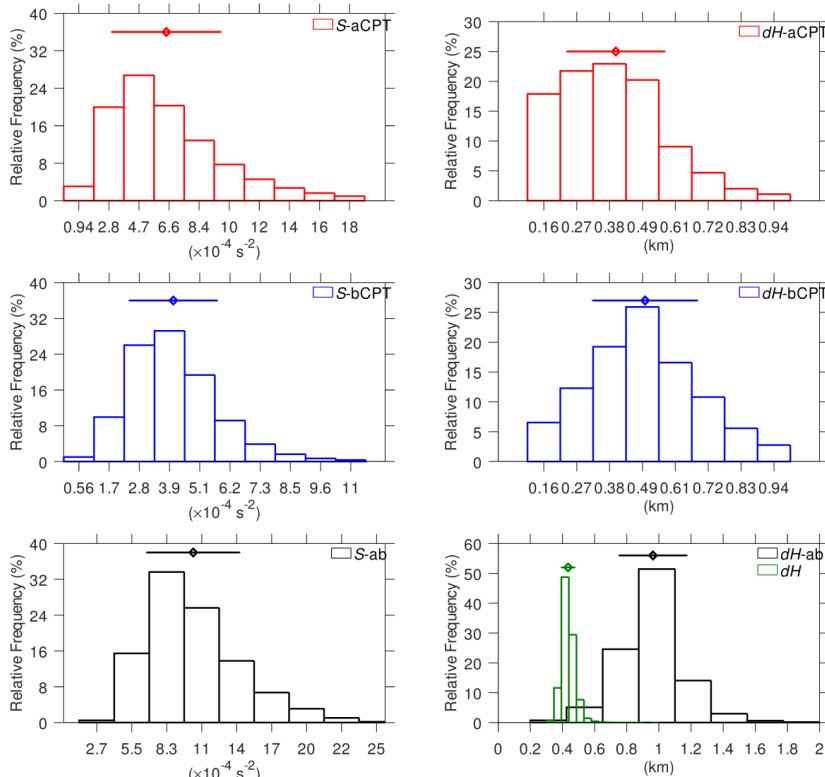

**Figure 9** Frequency distribution of TIL parameters with respect to the vertical distance from the CPT. The diamond and horizontal line above each histogram represent the mean and standard deviation of each parameter, respectively.





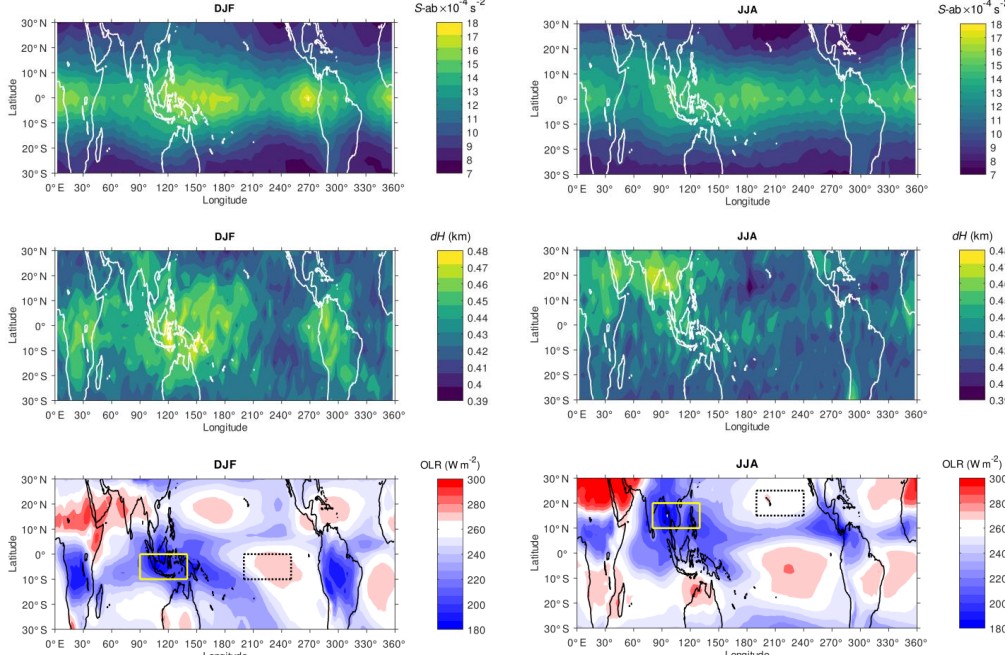

**Figure 10** Horizontal distribution of the mean *S*-ab (top row), *dH* (middle row), and OLR (bottom row) during Northern Hemisphere winter (DJF) and summer (JJA). Solid-yellow and dotted-black boxes in the bottom row indicate the sample "convective" and "non-convective" regions, respectively.



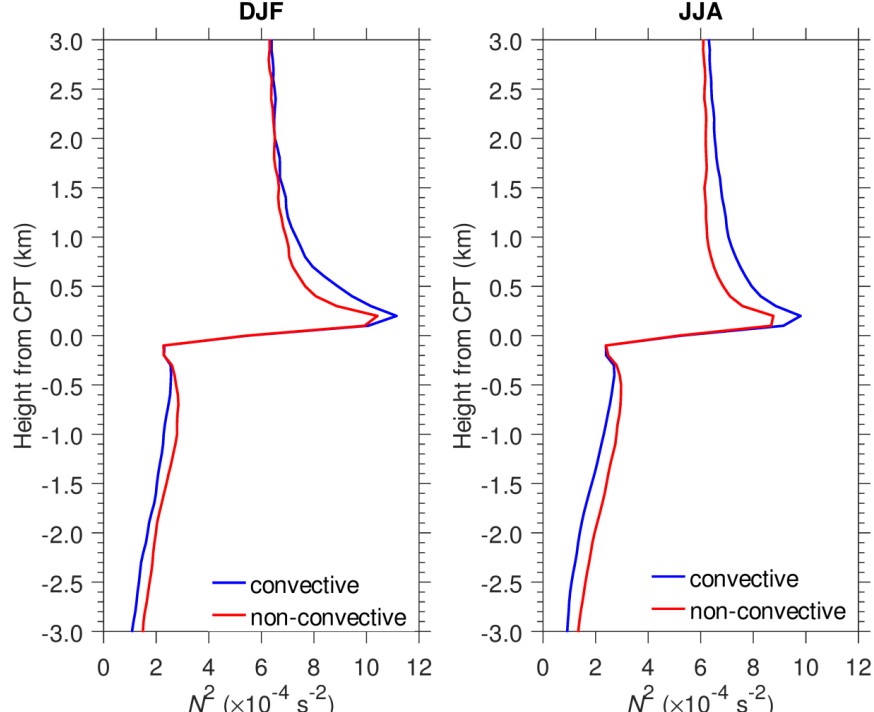

**Figure 11** Mean $N^2$ profiles over the convective and non-convective regions defined in Fig. 10 for DJF (left) and JJA (right).





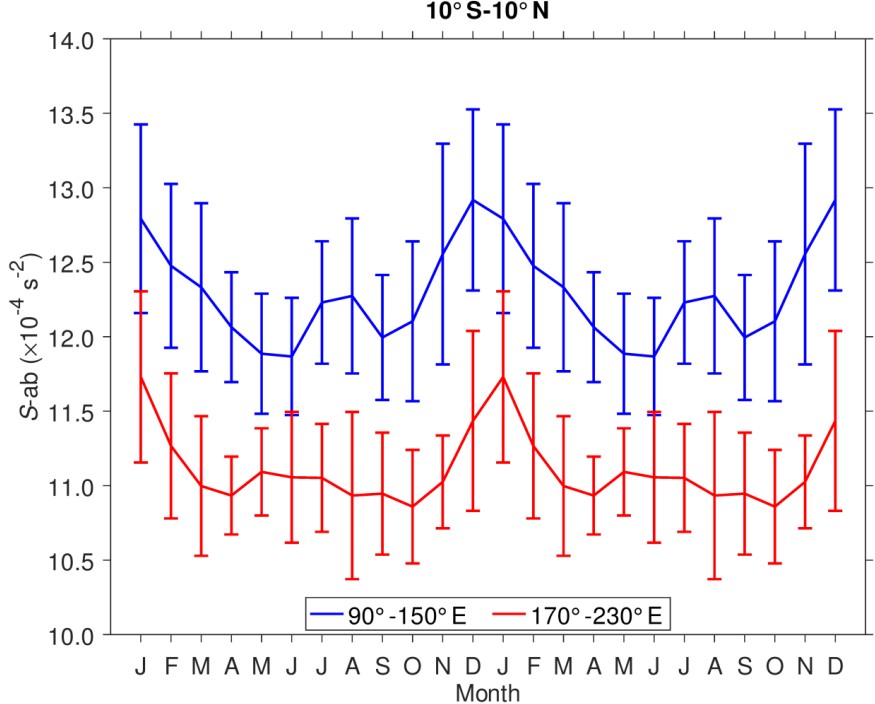

**Figure 12** Seasonal variation in the mean *S*-ab over the MC (90°–150°E) (blue line) and the PO (red line). Error bars show the standard deviation with respect to the mean value.



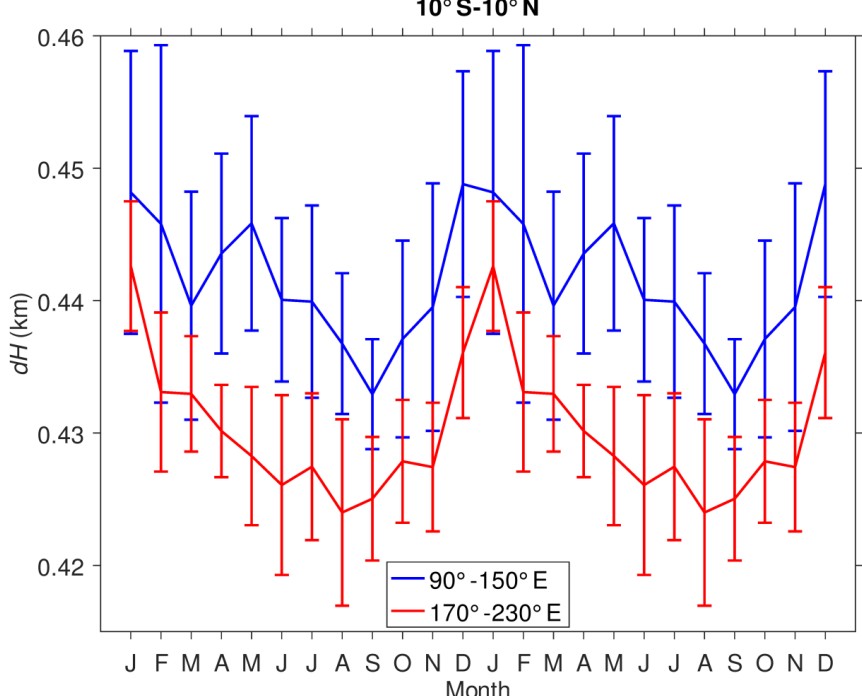

**Figure 13** As for Fig. 12, but for the mean and standard deviation of *dH*.



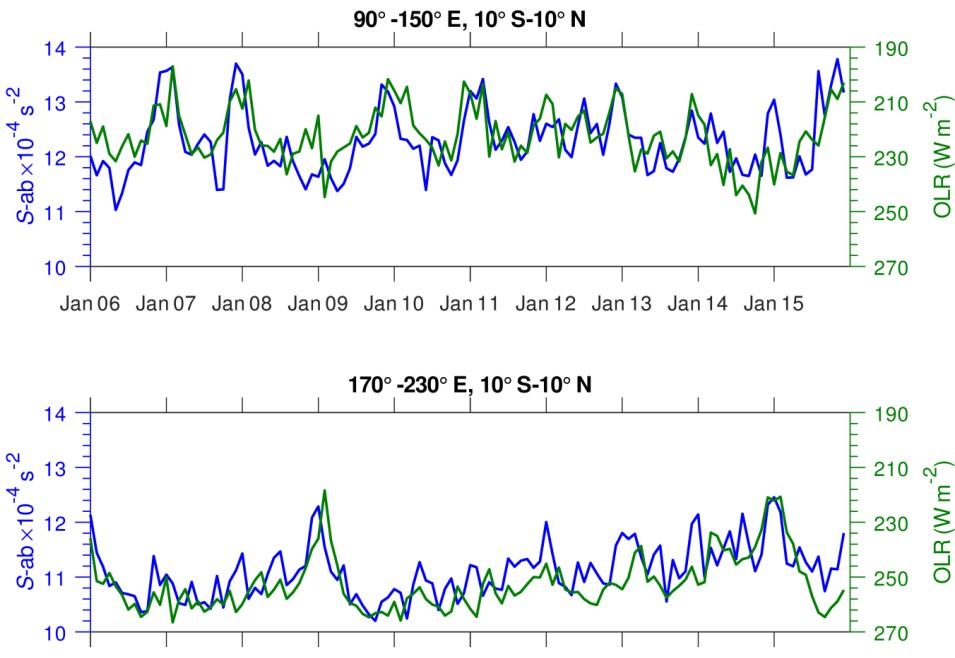

**Figure 14** Time series of monthly mean *S*-ab (blue) and OLR (green) for the MC (upper panel) and PO (lower panel).



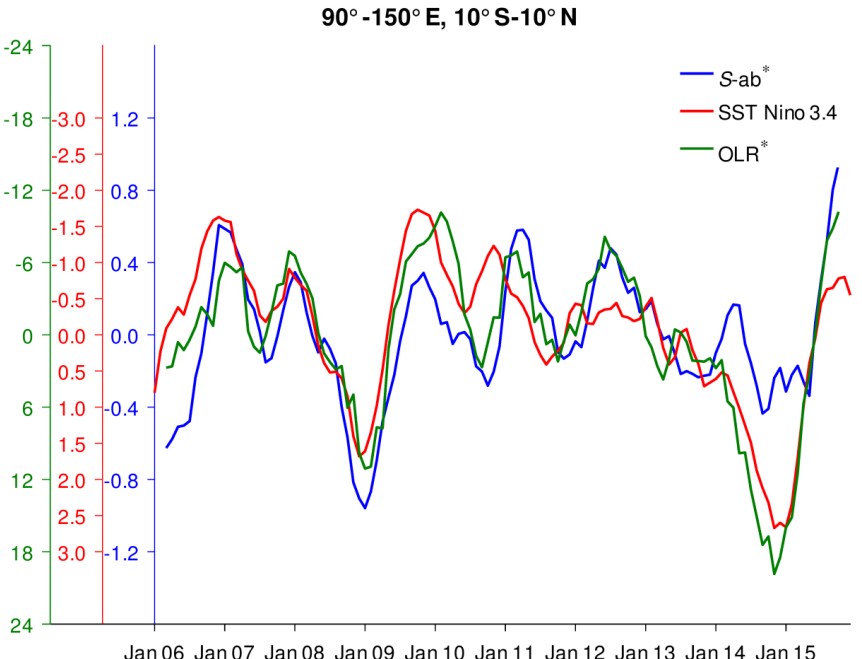

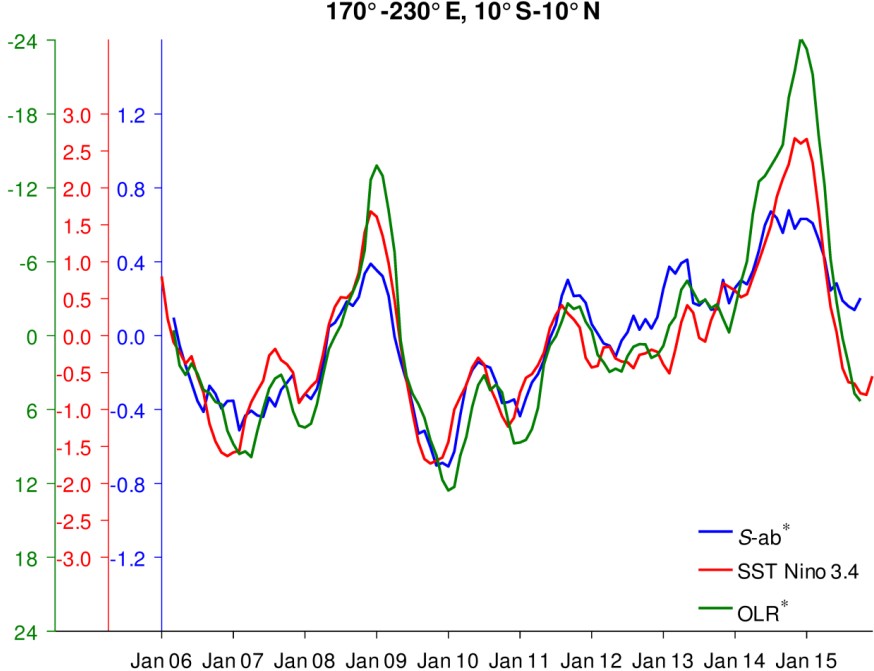

**Figure 15** Interannual variations in *S*-ab anomaly (*S*-ab*) (blue line), OLR anomaly (OLR*) (green line), and SST Nino3.4 index (red line)
5   for the MC (upper panel) and PO (lower panel). Note that the ordinate axes of both OLR* and SST Nino 3.4 index are reversed in the
upper panel, but only the ordinate axis of OLR* is reversed in the lower panel.



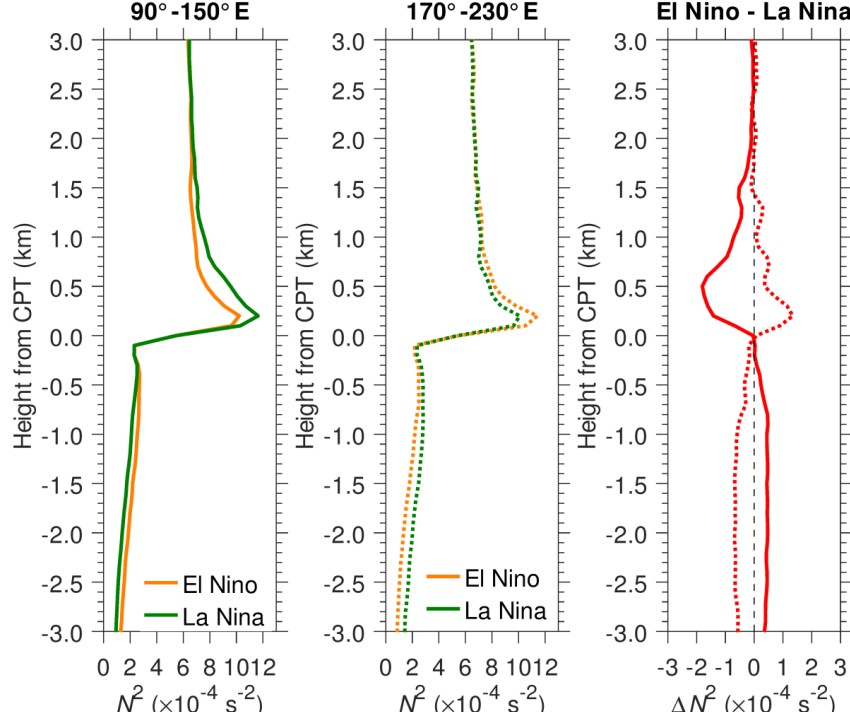

**Figure 16** Mean $N^2$ profiles during DJF El Niño (left) and DJF La Niña (middle). Difference between the mean $N^2$ in DJF El-Nino and in DJF La-Nina (right) in the MC (90°–150°E, solid line) and PO (170°–230°E, dotted line).





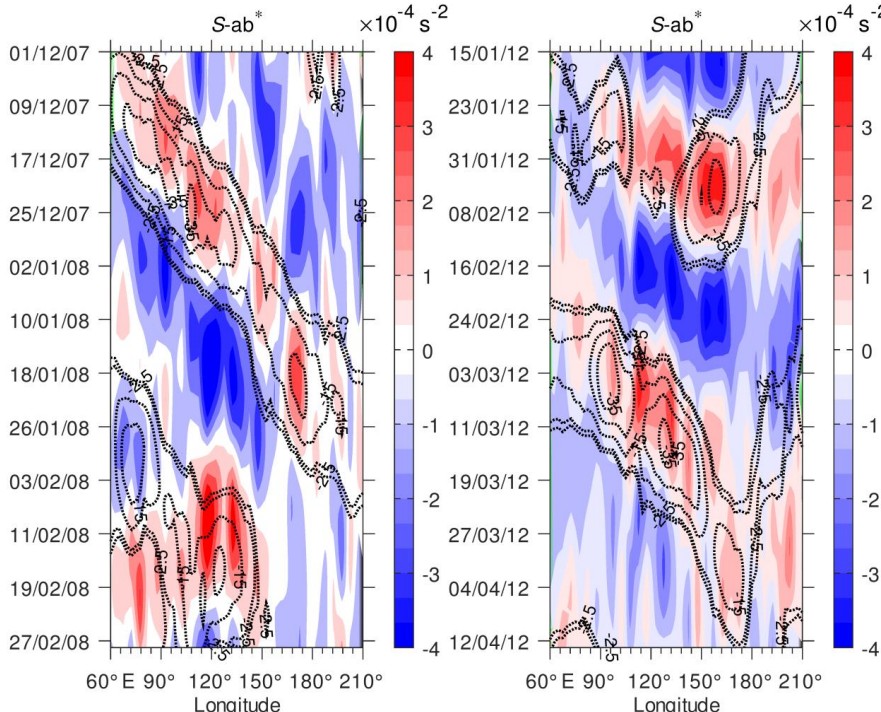

**Figure 17** Time–longitude distribution (Hovmöller diagram) of $S$-ab* (color shading/contours) and OLR* (dotted contours). Color contour interval is $0.25 \times 10^{-4}$ s$^{-2}$ and only negative values of OLR* are shown for $-35$, $-25$, $-15$, $-5$, and $-2.5$ W m$^{-2}$.



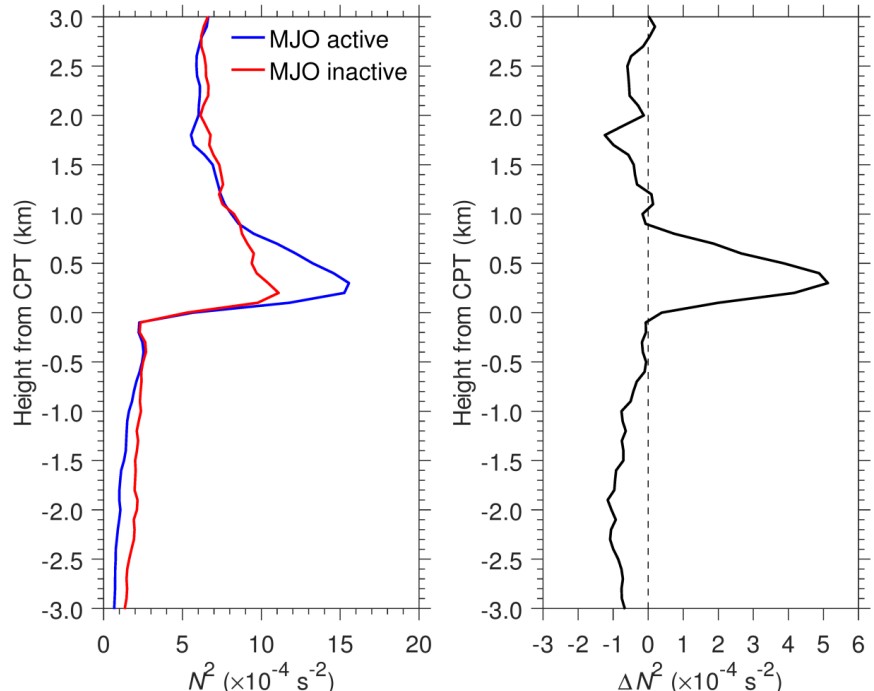

**Figure 18** Mean $N^2$ profiles during MJO active (blue) and MJO inactive (red) phases (left) and their difference (right).

