# Peer review of "Characteristics of the tropical tropopause inversion layer using highresolution temperature profiles retrieved from COSMIC GNSS Radio Occultation"

_Atmospheric Chemistry and Physics, 2018_

## Referee Comment (RC1) · Anonymous Referee #1 · 14 Jan 2019

This is a welcome study about the tropical TIL (a region where TIL literature is relatively sparse) and how ENSO and the MJO influence it. The manuscript is well organized and well written, presenting several novel results. I only see one important weakness before it can be published (see major comments 1-3), related to the amount of detail when the authors discuss their results, and how they fit/compare to previous works. Once this has been overcome, it will definitely be worthy of publication in ACP.

I hope the comments included below are helpful for this purpose.

[Figure]

**### Major comments**

**# 1**

Amount of detail in the result sections.

The text within your first result sections (3.1, 3.2, 3.3) could be shortened. Most of the results presented there basically agree with previous findings by Grise et al. (2010), Son et al. (2011), Kim and Son (2012), Pilch Kedzierski et al. (2016) or Randel et al. (2007) in their latitude or zonal structures and their seasonality. There are very little main results in these subsections that are really new, so one can move on quicker.

Things get way more interesting in sections 3.4 and 3.5, I see lots of novel material. However all is discussed in a hurry compared to the previous result sections. The discussion of the results in sections 3.4 and 3.5 should be extended, because this is the most important and novel part of your study. Related to this, see Major comment 2.

**# 2**

Referencing and highlighting what's new. Throughout the results sections I too often don't see what exactly is new and what agrees with previous studies. I'll go section by section here.

-Introduction:

p.2 l.11: more appropriate references are Birner (2006) and Grise et al. (2010), in the sense that they look at the TIL in a global sense, including the tropics. Perhaps also keep Birner et al. (2002) as the first one about the TIL, but the most relevant to introduce your study are the two from above.

p.2 l.15-20: Randel and Wu (2005) studied Kelvin waves from GPS-RO and how they affect the zonal structures of tropopause height and the surrounding T structures. Should be included among Tsuda et al. (1994).

p.2 l.40: I really miss references to Grise et al. (2010) within this paragraph.

-Paragraph in p.4 l.30, and paragraph in p.6 l.30: Grise et al. (2010) did a comparison of LRT and CPT - relative N2 profiles (Their Fig. 2). You should discuss and compare your results to theirs.

-Paragraph in p.6 l.40: I miss a discussion with Randel and Wu (2005), Kim and Son (2012) and Pilch Kedzierski et al. (2016) comparing your results to the modulation of the tropopause by Kelvin waves, MJO and other equatorial waves presented in those studies.

-p.7 l.30: you should compare your S-ab histograms to the ones of TP sharpness in Pilch Kedzierski et al. (2016). Although they use N2max there, this measure is comparable to yours since N2min below TP is always very low and N2max would dominate your distribution of S-ab.

-Section 3.3: discussion with Grise et al. (2010), Pilch Kedzierski et al. (2016), Son et al. (2011), Kim and Son (2012)... is completely missing. These studies show horizontal structures of TP sharpness and its seasonality. Also, note that S-ab is centered around the Equator, while your convective activity by OLR is not, so how can you leave out modulation by equatorial waves out of the discussion? (convectively coupled or not, the amplitudes of eq. waves by definition maximize there)

-Section 3.5: discuss your results comparing to Zeng et al. (2012), Kim and Son (2012) and Pilch Kedzierski et al. (2016), who all showed how MJO modulates the tropopause zonal structures, sharpness or T structure within the TTL with the use of COSMIC GPS-RO.

**# 3**

TIL sharpening by convection. Throughout the manuscript I find that discussions could be improved about how convection may sharpen the TIL. See Holloway and Neelin (2007), Paulik and Birner (2012) and Kim et al. (2018) for a detailed mechanism for tropopause cooling/sharpening by convection. A reference to these should be included

in your manuscript.

Also, I suggest to make a plot, for both PO and MC regions, showing a diagram of OLR versus S-ab of individual collocated RO profiles (e.g. in the same grid and day), similar to the diagram of Randel and Wu (2007) with rel. vorticity -vs- TP sharpness. In principle it should show increased S-ab with lower OLR values, at any region. With this you could link the convective influence on the TIL across different timescales (seasonal, MJO, ENSO) and it would be a great complement to figures 10-15 which are climatologies or monthly means.

**## Minor and technical corrections ## ## ##**

**Use of 'GNSS' throughout the manuscript: The term 'GNSS' is being used for recent satellite missions such as Metop (A, B and C), and for planning future missions. GNSS is the more general term which includes navigation satellites from all countries, while GPS is the American part. The idea is that GNSS receivers are able to capture signals from more satellites and yield more occultation profiles. Now, the Metop/GRAS instrument stands for 'GNSS Receiver for Atmospheric Sounding' while the IGOR instruments onboard COSMIC stand for 'Integrated GPS Occultation Receiver'. In Anthes et al. (2008) it is always referred to as GPS, the same in subsequent publications. So, for consistency, I see no reason not to use GPS in your manuscript.**

**Title: It's too general and needs to be more specific. I suggest to somehow highlight zonal structures and the influence of ENSO and the MJO in the title already. This is not the first manuscript to study the tropical TIL globally.**

**Abstract: First paragraph can be shortened: details about the resolution of your RO profiles or the definitions (S-ab, dH and so on) belong to the Data and Methods section. You can elaborate some more within the second paragraph, and simply use TIL sharpness and thickness instead of the acronyms there.**

**p.2**

l. 12: I think you mean '... very low temperature in the TTL...'

l. 23-25: This sentence is vague and difficult to follow. I suggest to formulate it this way: how ENSO modulates TTL temperature anomalies or wave activity.

l. 30: '(i.e. the sharpness)' doesn't fit there in the sentence. Rewrite.

**p.3**

l. 4: also mention the MJO and QBO in this sentence, their influence on the tropical TIL was analyzed in this study as well.

l. 6: also mention in this sentence that the real resolution of RO measurements increases in regions of increased refractivity gradients (such as inversion layers above the boundary layer or the tropopause), where it's most needed.

l.27: in 2015 and 2016 the number of profiles is significantly less than that.

**p.4 l.1: include a webpage here or within the Acknowledgements.**

**p.5**

l. 6-10: is it a simple mean, or do you apply any kind of weighing to get the grid's value?

l. 38: I'm confused by this sentence, wasn't your dataset always 0.1 km vertically resolved? Then how can your LRT be sensitive to vertical resolution?

**p.6 l. 16: I think you mean 'In agreement with previous studies'. Also, refer to those studies.**

**p.8 l.1-2: I'd erase this sentence, it's too vague.**

**p.10 l.10: this sentence is too speculative. Could as well be related to ENSO amplitudes within your study's time period, of only one decade.**

**p.11**

l. 21: I think statistics like 'x percent of values of this parameter are within y range'

are unnecessary in the Concluding remarks section, maybe even throughout the manuscript. I suggest instead to use a structure like: 'maxN2 is typically located within 0.5 km above the CPT' or 'typical dH values range within...' and refer to the corresponding figures, so that the important numbers are easier to digest for the reader.

l. 39: what is meant with 'from the new definitions'?

l. 40: as it reads now, this paragraph fits better in the introduction section as motivation or for discussions within your result sections. I suggest to remove it.

**## I also noticed some errors in your reference list: - First one is Andrews et al. (the 's' is currently missing throughout the manuscript) - The Anthes et al. reference for the COSMIC mission should rather be the one from 2008. - p.13 l.6: Kedzierski –> 'Pilch Kedzierski' (also throughout the manuscript)**

**Fig. 4: change colour of the blue line to something that contrasts more with the lower N2 values which are also blue.**

**Figs. 5, 6, 7, and 8: I really need to zoom a lot to see the features you're describing in the text. I suggest keeping the lower boundary at -1 km instead of -3, and removing the contour lines to leave only the color shading for better visibility of the values reached within the TIL.**

**Figs. 12 and 13: why are two repeated seasonal cycles displayed instead of one?**

**Fig. 14: could the authors provide a regression coefficient for these plots within the text?**

**##### References:**

Anthes, R. A., et al. (2008), The COSMIC/FORMOSAT-3 mission: Early results, Bull. Am. Meteorol. Soc., 89, 313, doi:10.1175/BAMS-89-3-313

Birner, T. (2006), Fine-scale structure of the extratropical tropopause region, J. Geophys. Res., 111, D04104, doi:10.1029/2005JD006301.

Holloway, C. E. and Neelin, J. D.: The convective cold top and quasi equilibrium, J. Atmos. Sci., 64, 1467–1487, 2007.

Kim, J., Randel, W. J., & Birner, T. (2018). Convectively driven tropopause-level cooling and its influences on stratospheric moisture. Journal of Geophysical Research: Atmospheres, 123, 590–606. https://doi.org/10.1002/2017JD027080

Paulik and Birner: Quantifying the deep convective temperature signal within the tropical tropopause layer (TTL), Atmos. Chem. Phys., 12, 12183–12195, 2012

Randel, W. J., and F. Wu (2005), Kelvin wave variability near the equatorial tropopause observed in GPS radio occultation measurements, J. Geophys. Res., 110, D03102, doi:10.1029/2004JD005006.

Zeng, Z., S.-P. Ho, S. Sokolovskiy, and Y.-H. Kuo (2012), Structural evolution of the Madden-Julian Oscillation from COSMIC radio occultation data, J. Geophys. Res., 117, D22108, doi:10.1029/2012JD017685.
* * *

---

## Referee Comment (RC2) · Anonymous Referee #2 · 18 Jan 2019

Comments to "Characteristics of the tropical tropopause inversion layer using high-resolution temperature profiles retrieved from COSMIC GNSS Radio Occultation" by Noersomadi Noersomadi et al.:

This study investigated the characteristics of the tropopause inversion layer using the high vertical resolution GNSS-RO data. It gave more details of the TIL sharpness and depth compared to previous studies, including the climatological mean and the intra-seasonal to interannual variations. In particular, this work gives a special focus on two different longitude regions (the Maritime Continent and the Pacific Ocean). It extended

previous work by related the interannual and intra-seasonal variations of the TIL to ENSO and MJO, respectively. It is well laid-out and well written. However, there are still some problems need to be fixed before publication.

Introduction The most relevant previous work to my understanding is Grise et al. 2010. Grise et al. 2010 already presented a global survey of the TIL strength (including annual cycle, horizontal distribution and interannual variations related to QBO) using the GPS-RO data with a vertical resolution also 0.1 km. However, it is not introduced in the introduction. It would be necessary to introduce results from Grise et al. 2010 and clearly describe what kind of improvement does this study want to have compared to them.

P2L12: "A very low temperature in the TIL", TTL should be better here

P3L6-10: I think it is now commonly known that the vertical resolution of the GPS/GNSS-RO data is up to 0.1 km. This sentence misleads the authors with a impression that the vertical resolution of GPS/GNSS-RO data is 1 km. Please rephrase this paragraph here.

Data P3L35-40: To my experience, the cosmic data from CDAAC should has data available for 0.1 km vertical resolution, e.g. wetPrf. Also note, most of studies as I know (Randel et al. 2007; 2010; Grise et al., 2010; Kedzierski et al., 2016 etc.) using cosmic data from the CDAAC and has vertical resolution of 0.1 km. I am not sure whether it is true that the cosmic2013 smoothed the data over a 0.5 km scale. Please check that carefully.

Figure 1: For the cosmic2013 data. The GPS-RO data is well known to be very accurate with high vertical resolution. From the Figure shown in Figure 1, the temperature profile is heavily smoothed. I strongly doubt for the results shown here. Please check carefully whether you are using the correct type of product from the cosmic2013 data.

TIL definition: There has been a lot of definitions to TIL strength, sharpness or

thickness. Beside the authors mentioned, please also include Randel et al. 2007; 2010 and Wang et al. 2013 for the TIL definition using the temperature gradient. Randel,W. J.,Wu, F., and Forster, P.: The Extratropical Tropopause Inversion Layer: Global Observations with GPS Data, and a Radiative Forcing Mechanism, J. Atmos. Sci., 64, 4489, doi:10.1175/2007JAS2412.1, 2007. Randel, W. J. and Wu, F.: The Polar Summer Tropopause Inversion Layer, J. Atmos. Sci., 67, 2572–2581, doi:10.1175/2010JAS3430.1, 2010. Wang,W., Matthes, K., Schmidt, T., and Neef, L.: Recent variability of the tropical tropopause inversion layer, Geophys. Res. Lett., 40, 6308–6313, doi:10.1002/2013GL058350, 2013.

Definition of the TIL thickness: I don't know whether it is necessary to have new definitions of TIL sharpness and thickness since there have been so many kinds of definitions. If the authors feel it is necessary, please address the reason clearly here. In particular, why a 80% is used for the dH. Are there any physical or statistical reasons? Otherwise, I would suggest to using the existed definitions, for example, the maximum of N2 above the tropopause for the TIL sharpness.

Results: For the global distribution and seasonal variations of the N2 and TIL, it would be very helpful to have some comparison to previous work like Grise et al. 2010. Please describe clearly whether the results are consistent with each other and what are new findings from the previous work.

Figures 5 and 7. The values of N2 above the tropopause is not clear. Please update the color map used for these figures.

---

## Referee Comment (RC3) · Anonymous Referee #3 · 10 Feb 2019

This manuscript presents global characteristics of static stability in different vertical coordinate systems and closely investigates its characteristics in the tropical tropopause region. The authors use ten years (2007 to 2016) of GNSS RO data from the COSMIC satellite constellation. Data from an FSI RO retrieval are used. These profiles have a much better vertical resolution than profiles from other RO processing centers. The tremendous vertical resolution of this data set is exploited by investigating atmospheric characteristics in detail. Spatial, inter-annual, annual, and intra-interannual variability of tropopause sharpness and the thickness of the tropopause inversion layer (TIL) are

discussed. In my opinion, this is an interesting study with important scientific results. To a large extend, the manuscript is clearly and concisely written. Exceptions are some parts of the introduction and data description (see minor comment below). More importantly, I have some doubts regarding the vertical resolution of the data set and the conclusions drawn (see major comment below).

**1  General comments**

1. The vertical resolution of the data set is specified to be 100 m. However, in some sections of the manuscript (mainly in section 3.3), the authors discuss characteristics with a distinctively better vertical resolution. Middle panels of Fig. 10 and also Fig. 13, e.g., show dH characteristics with a 10 m vertical resolution. I doubt somehow that this appropriate. Similarly, the authors find an uncertainty of the thickness of the TIL of 40 m. It is important to add a discussion and prove that these features can really be retrieved with this data set. See also minor comments on the retrieval and the uncertainty of the vertical grid.

**2  Minor comments**

- I have difficulties in following the logical structure of some parts of the introduction (section 1) and data description (section 2). More specifically, paragraphs 3, 4, and 5 of section 1 (pages 2 and 3), and several paragraphs of section 2 (i.e., the first paragraph of section 2, 2.1, and 2.2, the last paragraph of section 2.1 (concerning the TIL definition)) should be revised.

- Add some more detailed discussion about the data retrieval. What input data are used for the FSI retrieval? What is the vertical coordinate of your data set (height

above ellipsoid, height above geoid, geopotential height)? What is the uncertainty of the vertical grid? Scherllin-Pirscher et al. (2017, doi:10.1002/2016JD025902) also discuss some of these issues.

- Introduce all acronyms at their first occurrence.

- Abstract, line 22: It is not clear at this point what "S-ab anomaly (S-ab*)" and "OLR anomaly" refer to. Please rewrite this sentence.

- Introduction, page 2, paragraph 3: At least for a non-expert reader, the first two sentences "The vertical profile of $N^2$ across the tropopause (i.e., the sharpness) and the thickness of the layer of maximum $N^2$ above the tropopause have been determined in previous studies using both ground- and satellite-based observations. For example, Bell and Geller (2008) analyzed the twice daily standard radiosonde data from the WMO stations and found that the thickness was ~1 km at low latitudes." are not clear. Does "the thickness" refer to the layer between the tropopause and maximum $N^2$ above the tropopause? Please clarify.

- Introduction, page 2, paragraph 4: the better reference for the COSMIC data set is Anthes et al. (2008, doi:10.1175.BAMS-89-3-313).

- Introduction, page 3, paragraph 2: Add the Noersomadi and Tsuda (2017)-reference.

- Section 2.1, page 3, line 27: As far as I know, COSMIC does not provide 1500 to 2000 profiles anymore. Please clarify.

- Section 2.1 page 4, lines 6/7: I do not understand the explanation "caused by different truncations of the GNSS signals in the lower atmosphere". It is true that the penetration depth is different for each RO measurement. It is also true that the penetration depths of the retrieved profiles can be different for different RO retrievals. However, I do not understand the connection between penetration depth

and the number of profiles. Is there a specific quality indicator which reduces the number of profiles for cosmicfsi? Please clarify.

- Section 2.1, page 4, lines 15/16: "which is located within 115 km horizontal radius": Does this number refer to the mean tangent point location? How is it defined? Did you account for the tangent point drift? Please explain.

- Section 2.2, page 4, line 36 to page 5, line 5: I recommend referring to the right panel of Fig. 2 for this explanation. Furthermore, the figure should include all parameters introduced in this section.

- Section 3.1, page 5, lines 33/34: "Figure 3a-d...": This sentence is a general statement, which should be made earlier in this paragraph.

- Section 3.1, page 6, line 1: "the LRT should be the same as the lowest CPT" and line 40: "the LRT and CPT are at the same location/height". In the tropics, LRT is usually lower than CPT because LRT refers to a specific temperature gradient and CPT to the temperature minimum. Please clarify.

- Section 3.1, page 6, lines 39 to 42: I do not understand this explanation. Please rewrite.

- Section 3.1, page 7, lines 15/16: I do not understand the sentence "The level of increasing temperature can be determined as the LRT height when the WMO definition is attained" as the LRT is not determined by increasing temperatures but only a temperature gradient threshold.

- Section 3.2, page 7, Figure 9: The class widths of several panels of Fig. 9 seem to be either non-equidistant or floating numbers causing rounding errors. Please use well defined and equidistant classes.

- Section 3.2, page 7, lines 28/29: Since Fig. 9 shows that mean S-aCPT is smaller than $6.6 \times 10^{-4}$ s$^{-2}$ this statement cannot be true. Please clarify.

- Section 3.3, page 8, lines 13/14: "The highest values, up to $16 - 18 \times 10^{-4}$ s$^{-2}$, are associated with low OLR values...": In DJF, S-ab is clearly above average ($17 - 18 \times 10^{-4}$ s$^{-2}$) west of South America, where OLR values are about 260 W/m$^2$, defined as "non-convective" on page 8, lines 33/34. Above the convective regions in South America, however, where OLR values are really low, the S-ab only reaches about $14 \times 10^{-4}$ s$^{-2}$. So this statement seems to be wrong for the South American region. Please clarify.

- Section 3.4, page 9, line 22: How big are the cross-correlations between S-ab and OLR at different lags?

**3  Figures**

Note there are also some other comments above.

- I recommend adding a background grid in Figs. 1, 2, 11, 12, 13, 14, 16, 18

- Please also add minor ticks on x-axis of Figs. 11, 14, right panel of Fig. 18, and on y-axis of Figs. 12, 13, 16

- I recommend adding "MC" and "PO" next to the longitudinal information in the figure titles of Figs. 3, 5, 6, 7, 14, 15, 16.

- Please indicate MC, PO, Atlantic, and Indian Ocean in Figs. 4, 7, and 8 (e.g., arrows below the x-axis).

- Indicate MC and PO in the figure legends of Figs. 12, 13 (could be one two panel figure) and also in the right panel of Fig. 16.

- I suggest indicating the LRT and CPT for all data sets in Fig. 1

- Indicate El Nino events 2008/2009 and 2014/2015 in both panels of Figs. 14 and 15.

**4 Editorial**

- Page 1, line 15: "in the range" → "in the range of"

- Page 3, line 8: "Centre" → "Center"

- Page 3, line 17: "cosmic2013 only" → "cosmic2013 data only"

- Page 4, line 5: "within a 183-day period" → "within the 183-day period in 2011/2012".

- Page 4, line 16: According to Table 1, Surabaya station is located at 112.78°E, 7.37°S. Please clarify.

- Page 4, line 18: "less than 30 min": This is only true for the UTLS.

- Page 4, line 21: "within a 183-day period" → "within the 183-day period".

- Page 6, line 4: "height-longitude" → "longitude-height"

- Page 6, line 10: "height-latitude" → "latitude-height"

- Page 6, line 30: "Fig. 6b" → "Fig. 6c"

- Page 7, line 4: "height-longitude" → "longitude-height"

- Page 7, line 19: "than in JJA." → "than in JJA (Fig. 8)".

- Page 7, line 27: "entire latitude range" → "entire region from"

- Page 10, line 27: "15 April 2012" might be "13 April 2012"

---

## Author Comment (AC1) · 25 Mar 2019

Reply to the comments by the Reviewer #1

This is a welcome study about the tropical TIL (a region where TIL literature is relatively sparse) and how ENSO and the MJO influence it. The manuscript is well organized and well written, presenting several novel results. I only see one important weakness before it can be published (see major comments 1-3), related to the amount of detail when the authors discuss their results, and how they fit/compare to previous works.

Once this has been overcome, it will definitely be worthy of publication in ACP. I hope the comments included below are helpful for this purpose.

We deeply appreciate the reviewer for providing constructive comments and suggestions to our manuscript. We show below our responses to the individual comments.

**### Major comments ### ## 1 # Amount of detail in the result sections. The text within your first result sections (3.1, 3.2, 3.3) could be shortened. Most of the results presented there basically agree with previous findings by Grise et al. (2010), Son et al. (2011), Kim and Son (2012), Pilch Kedzierski et al. (2016) or Randel et al. (2007) in their latitude or zonal structures and their seasonality. There are very little main results in these subsections that are really new, so one can move on quicker. Things get way more interesting in sections 3.4 and 3.5, I see lots of novel material. However all is discussed in a hurry compared to the previous result sections. The discussion of the results in sections 3.4 and 3.5 should be extended, because this is the most important and novel part of your study. Related to this, see Major comment 2.**

We agree with the reviewer. We have shortened the text particularly section 3.1 by focusing on the N2 distribution in the tropics. We show latitude-height cross section of N2 during the two seasons both in the MC and PO and longitude-height cross section relative to CPT. We have modified sections 3.2 and 3.3 by following the suggestions by the reviewer (see below for the details). We have extended discussion in sections 3.4 and 3.5, adding the diagram of S-ab versus OLR in section 3.5 (see also below for the details).

**# 2 # Referencing and highlighting what's new. Throughout the results sections I too often don't see what exactly is new and what agrees with previous studies. I'll go section by section here.**

-Introduction: p.2 l.11: more appropriate references are Birner (2006) and Grise et al. (2010), in the sense that they look at the TIL in a global sense, including the tropics. Perhaps also keep Birner et al. (2002) as the first one about the TIL, but the most

relevant to introduce your study are the two from above. We add the following statements in P2 L12-16: "Using routine radiosonde sounding data, a strong mean inversion at the tropopause in the midlatitude was analyzed (Birner et al., 2002; Birner, 2006). Grise et al. (2010) conducted a global survey of the TIL characteristics, including annual cycle, horizontal distribution and interannual variations related to the stratospheric Quasi Biennial Oscillation (QBO) using the Global Positioning System Radio Occultation (GPS-RO) data." We also add the statements citing Birner (2006) and Grise et al. (2010) in P2 L23-26. "Birner (2006) found that the mean N2 shows enhanced values near the extratropical tropopause compared to the extratropical lower stratosphere. Furthermore, Grise et al. (2010) found the largest magnitudes of N2 between $10°-15°$ latitude in both hemisphere during northern hemisphere (NH) winter season."

p.2 l.15-20: Randel and Wu (2005) studied Kelvin waves from GPS-RO and how they affect the zonal structures of tropopause height and the surrounding T structures. Should be included among Tsuda et al. (1994). We follow suggestion by the reviewer and add the following statement in P2 L32-33: "Randel and Wu (2005) demonstrated eastward phase tilt with height of Kelvin waves that modulate the climatological cold tropopause over Indonesia with the maximum amplitude near the tropical tropopause ($\sim$17 km)."

p.2 l.40: I really miss references to Grise et al. (2010) within this paragraph. We add Grise et al. (2010) following suggestion by the reviewer in P3 L17.

-Paragraph in p.4 l.30, and paragraph in p.6 l.30: Grise et al. (2010) did a comparison of LRT and CPT - relative N2 profiles (Their Fig. 2). You should discuss and compare your results to theirs. We add the following sentences in P6 L23-26: "We found the maxN2 at range $11-12.0 \times 10-4$ s$-2$ above CPT height in the two longitude regions. Note that Grise et al. (2010) reported that the zonal mean maxN2 value above the LRT and CPT heights was $\sim$8.0 $\times$ 10$-4$ s$-2$ using the CHAMP dataset (see fig.2 in Grise et al., 2010). The different larger values found in this study are probably due to the use of data of higher effective vertical resolution."

-Paragraph in p.6 l.40: I miss a discussion with Randel and Wu (2005), Kim and Son (2012) and Pilch Kedzierski et al. (2016) comparing your results to the modulation of the tropopause by Kelvin waves, MJO and other equatorial waves presented in those studies. We add the following sentences in P6 L30-34: "The results shown in Fig. 3 uncover the detail structure of N2 above CPT in the specific longitude regions compared to the results by Grise et al. (2010) that showed the mean N2 over $0-1$ km layer above LRT. The vertical propagation of equatorial waves (Kelvin waves and/or gravity waves), as the results of convective forcing, modulates the tropopause (Tsuda et al., 1994; Randel and Wu, 2005; Kim and Alexander, 2015; Kim et al., 2018). The MJO activity was also found to control the tropopause variability (Kim and Son, 2012; Pilch Kedzierski et al., 2016)."

-p.7 l.30: you should compare your S-ab histograms to the ones of TP sharpness in Pilch Kedzierski et al. (2016). Although they use N2max there, this measure is comparable to yours since N2min below TP is always very low and N2max would dominate your distribution of S-ab. We add the following sentences in P7 L16-19: "The positive skewness distribution of S-ab is found to be similar to the results by Pilch Kedzierski et al. (2016) for maxN2+1 above LRT for both easterly and westerly QBO periods. This is reasonable since maxN2+1 above CPT would dominate the relatively low values of minN2$-$1 below CPT, showing similar features to those of S-ab."

-Section 3.3: discussion with Grise et al. (2010), Pilch Kedzierski et al. (2016), Son et al. (2011), Kim and Son (2012)... is completely missing. These studies show horizontal structures of TP sharpness and its seasonality. Also, note that S-ab is centered around the Equator, while your convective activity by OLR is not, so how can you leave out modulation by equatorial waves out of the discussion? (convectively coupled or not, the amplitudes of eq. waves by definition maximize there) We add the following statements in P7 L41 – P8 L6: "Large S-ab values are found along in the equator region, while low OLR regions show latitudinal variation with season. Local and seasonal variability of horizontal structure of tropopause sharpness presented in this work is consistent with

previous studies which attributed it to equatorial waves activity (e.g. Grise et al., 2010; Son et al., 2011; Kim and Son, 2012). However, we found different quantitative results in particular over the Western Pacific because of the use of maxN2+1 and minN2–1 instead of averaging N2 within ±1 km relative to CPT by Kim and Son (2012), and also because of the use of data of higher effective vertical resolution. Maximum static stability just above the tropical tropopause could also be associated with divergence flow as demonstrated by Pilch Kedzierski et al. (2016)."

-Section 3.5: discuss your results comparing to Zeng et al. (2012), Kim and Son (2012) and Pilch Kedzierski et al. (2016), who all showed how MJO modulates the tropopause zonal structures, sharpness or T structure within the TTL with the use of COSMIC GPS-RO. We add the following sentences in P10 L30-36: "Zeng et al. (2012) demonstrated cold temperature anomaly observed with COSMIC near the tropopause associated with rainfall anomaly during MJO active phase propagation. Kim and Son (2012) and Pilch Kedzierski et al. (2016) also found MJO signal modulates the tropopause temperature and sharpness structure using COSMIC temperature data. We found that the MJO propagation in the tropics has an impact on the variability in tropopause sharpness, being consistent with the results by Pilch Kedzierski et al. (2016). Note that, we demonstrated time evolution of positive sharpness anomaly associated with negative OLR anomaly, which is not shown in the previous studies (Kim and Son, 2012; Pilch Kedzierski et al., 2016)."

**# 3**

TIL sharpening by convection. Throughout the manuscript I find that discussions could be improved about how convection may sharpen the TIL. See Holloway and Neelin (2007), Paulik and Birner (2012) and Kim et al. (2018) for a detailed mechanism for tropopause cooling/sharpening by convection. A reference to these should be included in your manuscript. We add the sentences in P11 L2 – 7: It is well known that vertical structure of temperature perturbations associated with deep convection indicate warm anomaly in the middle and upper troposphere and cold anomaly near the tropopause

(e.g. Holloway and Neelin, 2007; Paulik and Birner, 2012). Adiabatic cooling near the tropopause is a natural response to the diabatic warming due to convection as a result of hydrostatic adjustment (Holloway and Neelin, 2007). Kim et al. (2018) hypothesized that cold anomaly near the tropopause associated with organized deep convection during the MJO active phase is due to dehydration processes.

We also add some statements in the introduction referring to the paper suggested by the reviewer in P2 L42 – P3 L4: "Holloway and Neelin (2007) found negative correlation of temperature perturbations between the free troposphere (about 800 – 200 hPa) and the convective cold top (about 100 – 50 hPa). The cooling in the convective cold top is due to a hydrostatic adjustment to the deep convective heating (Holloway and Neelin, 2007). Paulik and Birner (2012) identified typical temperature perturbations in the deep convective cloud associated with reduced ozone event, warm anomaly in the mid and upper troposphere, and cold anomaly in the tropical tropopause layer (TTL). We are also interested in the characteristics of static stability in the TTL associated with the convective activity."

Also, I suggest to make a plot, for both PO and MC regions, showing a diagram of OLR versus S-ab of individual collocated RO profiles (e.g. in the same grid and day), similar to the diagram of Randel and Wu (2007) with rel. vorticity -vs- TP sharpness. In principle it should show increased S-ab with lower OLR values, at any region. With this you could link the convective influence on the TIL across different timescales (seasonal, MJO, ENSO) and it would be a great complement to figures 10-15 which are climatologies or monthly means. Thank you very much for your suggestion. We have created the following diagrams of OLR versus S-ab as Fig.14 for the revised manuscript, and add the following sentences in P11 L10-17.

Figure 14 Diagrams of S-ab versus OLR (top), and of the number of samples for S-ab and OLR (middle and bottom, respectively) in the MC (left) and PO (right) regions.

The added text: "To show the relationship between the convective activity and

tropopause sharpness in the MC and PO regions at intra-seasonal through to inter-annual time scales, we provide the diagrams in Fig. 14 which are analogous to fig. 3 of Pilch Kedzierski et al. (2016) that showed sharpness versus divergence near the tropopause, and to fig. 5 of Randel et al. (2007) that showed sharpness versus vorticity in the extratropics. Fig. 14 displays the scatter plot of S-ab versus OLR in all seasons. As expected, large S-ab values at range $9-18 \times 10-4$ s$-2$ are associated with OLR values of $250-150$ W m$-2$ for the number of samples > 180. We have also investigated the slope of relationship between S-ab and OLR in the top panel of Fig. 14 for the number of samples $\geq 500$. Both slopes show only slight difference between the MC and PO regions, indicating that the variability of deep convection is the major cause of tropopause sharpness variability."

**## Minor and technical corrections ## ## ##**

**Use of 'GNSS' throughout the manuscript: The term 'GNSS' is being used for recent satellite missions such as Metop (A, B and C), and for planning future missions. GNSS is the more general term which includes navigation satellites from all countries, while GPS is the American part. The idea is that GNSS receivers are able to capture signals from more satellites and yield more occultation profiles. Now, the Metop/GRAS instrument stands for 'GNSS Receiver for Atmospheric Sounding' while the IGOR instruments onboard COSMIC stand for 'Integrated GPS Occultation Receiver'. In Anthes et al. (2008) it is always referred to as GPS, the same in subsequent publications. So, for consistency, I see no reason not to use GPS in your manuscript. We agree with the reviewer and use the term GPS in the revised manuscript.**

**Title: It's too general and needs to be more specific. I suggest to somehow highlight zonal structures and the influence of ENSO and the MJO in the title already. This is not the first manuscript to study the tropical TIL globally. We appreciate your suggestion and modify the title as: Influence of ENSO and MJO on the zonal structure of tropical tropopause inversion layer using high-resolution temperature profiles retrieved from COSMIC GPS Radio Occultation**

**Abstract: First paragraph can be shortened: details about the resolution of your RO profiles or the definitions (S-ab, dH and so on) belong to the Data and Methods section. You can elaborate some more within the second paragraph, and simply use TIL sharpness and thickness instead of the acronyms there. We would like to keep the details about the resolution because that is very important information of our COSMIC retrieval. Grise et al. (2010) used CHAMP dataset which freely available from the UCAR. On the other hand, Noersomadi and Tsuda (2017) discussed two GPS-RO products by the UCAR (i.e. COSMIC version 2010 and re-processed data version 2013). Noersomadi and Tsuda (2017) showed that COSMIC version 2010 indicated mixture of vertical resolution (0.1 – 1 km), while COSMIC version 2013 fixed the resolution at 0.5 km in the UTLS (see their result in fig. 3). Details explanation about the retrieval of COSMIC re-processed data version 2013 can be found in Zeng et al (2016).**

**p.2**

l. 12: I think you mean '... very low temperature in the TTL...' We have revised as suggested.

l. 23-25: This sentence is vague and difficult to follow. I suggest to formulate it this way: how ENSO modulates TTL temperature anomalies or wave activity. We have modified the sentence in P2 L35-37 as: "The El Niño and La Niña events, known as the El Niño Southern Oscillation (ENSO), considerably influence the equatorial wave activity and the TTL structure (Trenberth, 1997; Nishimoto and Shiotani, 2012, Scherllin-Pirscher et al., 2012)."

l. 30: '(i.e. the sharpness)' doesn't fit there in the sentence. Rewrite. We refer to Fig. 2 (right panel) for clarity (P3 L6-8).

**p.3**

l. 4: also mention the MJO and QBO in this sentence, their influence on the tropical TIL was analyzed in this study as well. We have revised as suggested in P3 L23.

[Figure]

l. 6: also mention in this sentence that the real resolution of RO measurements increases in regions of increased refractivity gradients (such as inversion layers above the boundary layer or the tropopause), where it's most needed. We have revised as suggested in P3 L28-29.

l.27: in 2015 and 2016 the number of profiles is significantly less than that. We add the following statement in P4 L10-11: "Nevertheless, the number of profiles is significantly decreasing in 2015 and 2016 ($60-100$ profiles over $10°$S–$10°$N)."

**p.4 l.1: include a webpage here or within the Acknowledgements. We have added the webpage information in the Acknowledgements.**

**p.5**

l. 6-10: is it a simple mean, or do you apply any kind of weighing to get the grid's value? We add the following statement in P6 L1: "... using simple arithmetic mean ..."

l. 38: I'm confused by this sentence, wasn't your dataset always 0.1 km vertically resolved? Then how can your LRT be sensitive to vertical resolution? We remove this statement because we do not show the N2 profile relative to LRT.

**p.6 l. 16: I think you mean 'In agreement with previous studies'. Also, refer to those studies. We remove this statement because we do not show the N2 profile relative to LRT.**

**p.8 l.1-2: I'd erase this sentence, it's too vague. We have removed this as suggested.**

**p.10 l.10: this sentence is too speculative. Could as well be related to ENSO amplitudes within your study's time period, of only one decade. We add the following sentences in P10 L2 – 6: "We found that above the CPT the peak amplitude $\Delta$N2 in the MC region is higher and larger (0.5 km above the CPT and $1.8 \times 10-4$ s$-2$) than in the PO region (0.2 km above CPT and $1.4 \times 10-4$ s$-2$). The difference in the thermal structure near tropopause over land and ocean may cause the difference in peak amplitude of $\Delta$N2 above the CPT. One possibility is that the mountainous characteristics of the MC region are favorable to generate convectively couple equatorial waves (Kubokawa et al., 2016)."**

**p.11**

l. 21: I think statistics like 'x percent of values of this parameter are within y range' are unnecessary in the Concluding remarks section, maybe even throughout the manuscript. I suggest instead to use a structure like: 'maxN2 is typically located within 0.5 km above the CPT' or 'typical dH values range within...' and refer to the corresponding figures, so that the important numbers are easier to digest for the reader. We have revised as suggested.

l. 39: what is meant with 'from the new definitions'? We reply to this comment together with the next comment.

l. 40: as it reads now, this paragraph fits better in the introduction section as motivation or for discussions within your result sections. I suggest to remove it. We agree with the reviewer and remove this paragraph. We add the following sentences referring to Fig. 14 in P12 L4-6: "The diagram of S-ab versus OLR both in the MC and PO regions during all seasons indicates that the variability of convective activity in the tropics is the major cause of that of the tropopause structure at various time scales."

**## I also noticed some errors in your reference list: - First one is Andrews et al. (the 's' is currently missing throughout the manuscript) We have corrected this.**

- The Anthes et al. reference for the COSMIC mission should rather be the one from 2008. We have corrected this.

- p.13 l.6: Kedzierski –> 'Pilch Kedzierski' (also throughout the manuscript) We have corrected this.

**Fig. 4: change colour of the blue line to something that contrasts more with the lower N2 values which are also blue. We do not include this figure in the revised manuscript and we focus on the N2 relative to CPT.**

**Figs. 5, 6, 7, and 8: I really need to zoom a lot to see the features you're describing in the text. I suggest keeping the lower boundary at -1 km instead of -3, and removing the contour lines to leave only the color shading for better visibility of the values reached within the TIL. We have modified the figures as suggested.**

**Figs. 12 and 13: why are two repeated seasonal cycles displayed instead of one? We keep these figures in Fig.8 after removing some figures, and then add the following statement in P8 L40: "We repeat two seasonal cycles in Fig. 8 to display seasonal cycle more clearly."**

**Fig. 14: could the authors provide a regression coefficient for these plots within the text? We have added correlation coefficient in the figure title following suggestion by the reviewer. The figure number changes into Fig. 9.**

Thank you very much again for your very valuable comments and suggestions.

Reference Birner, T., Dörnbrack, A., and Schumann, U.: How sharp is the tropopause at midlatitudes?, Geophys. Res. Lett., 29, 1700, doi:10.1029/2002GL015142, 2002. Birner, T.: Fine-scale structure of the extratropical tropopause region, J. Geophys. Res., 111, D04104, doi:10.1029/2005JD006301, 2006. Gettelman, A., and Wang, T.: Structural diagnostics of the tropopause inversion layer and its evolution, J. Geophys. Res., 120, 46–62, doi:10.1002/2014JD021846, 2015. Grise, K. M., Thompson, D. W. J., and Birner, T.: A global survey of static stability in the stratosphere and upper troposphere, J. Clim., 23, 2275–2292, doi:10.1175/2009JCLI3369.1, 2010. Holloway, C. E., & Neelin, J. D.: The convective cold top and quasi equilibrium. Journal of the Atmospheric Sciences, 64(5),1467–1487, https://doi.org/10.1175/JAS3907.1, 2007. Kedzierski, R. P., Matthes, K. and Bumke, K.: The tropical tropopause inversion layer: Variability and modulation by equatorial waves, Atmos. Chem. Phys., 16, 11617–11633, doi:10.5194/acp-16-11617-2016, 2016. Kim, J.-E., and Alexander, M. J.: Direct impacts of waves on tropical cold point tropopause temperature, Geophys. Res. Lett., 42, 1584–1592, doi:10.1002/2014GL062737, 2015. Kim, J. and Son, S.

[Figure]
Interactive comment

-W.: Tropical cold-point tropopause: Climatology, seasonal cycle, and intraseasonal variability derived from COSMIC GPS radio occultation measurements, J. Clim., 25, 5343–5360, doi:10.1175/JCLI-D-11-00554.1, 2012. Nishimoto, E., and Shiotani, M.: Seasonal and interannual variability in the temperature structure around the tropical tropopause and its relationship with convective activities, J. Geophys. Res., 117, 1–11, doi:10.1029/2011JD016936, 2012. Noersomadi and Tsuda, T.: Comparison of three retrievals of COSMIC GPS radio occultation results in the tropical upper troposphere and lower stratosphere, Earth Planets Space., 69, doi:10.1186/s40623-017-0710-7, 2017. Paulik, L. C., & Birner, T.: Quantifying the deep convective temperature signal within the tropical tropopause layer (TTL). Atmospheric Chemistry and Physics, 12(24), 12,183–12,195. https://doi.org/10.5194/acp-12-12183-2012, 2012. Randel, W. J., & Wu, F.: Kelvin wave variability near the equatorial tropopause observed in GPS radio occultation measurements, Journal of Geophysical Research, 110, D03102. https://doi.org/10.1029/2004JD005006, 2005. Sokolovskiy, S., Schreiner, W., Zeng, Z., Hunt, D., Kuo, Y. -H, Meehan, T. K., Stecheson, T.W., Manucci, A.J., Ao, C.O.: Use of the L2C signal for inversions of GPS radio occultation data in the neutral atmosphere, GPS Solut, 18, 404−416, doi:10.1007/s10291-013-0340-x, 2014. Tsuda, T., Murayama, Y., Wiryosumarto, H., Harijono, S. W. B., and Kato, S.: Radiosonde observations of equatorial atmosphere dynamics over Indonesia: 1. Equatorial waves and diurnal tides, J. Geophys. Res., 99, 10491–10505, doi:10.1029/94JD00355, 1994. Zeng, Z., S.-P. Ho, S. Sokolovskiy, and Y.-H. Kuo: Structural evolution of the Madden-Julian Oscillation from COSMIC radio occultation data, J. Geophys. Res., 117, D22108, doi:10.1029/2012JD017685, 2012. Zeng, Z., Sokolovskiy, S., Schreiner, W., Hunt, D., Lin, J. and Kuo, Y. H.: Ionospheric correction of GPS radio occultation data in the troposphere, Atmos. Meas. Tech., 9, 335–346, doi:10.5194/amt-9-335-2016, 2016.

Please also note the supplement to this comment:
https://www.atmos-chem-phys-discuss.net/acp-2018-1182/acp-2018-1182-AC1-supplement.pdf

[Figure]

**Fig. 1.** Diagrams of S-ab versus OLR (top), and of the number of samples for S-ab and OLR (middle and bottom, respectively) in the MC (left) and PO (right) regions.

---

## Author Comment (AC2) · 25 Mar 2019

Reply to the comments by the Reviewer #2

Comments to "Characteristics of the tropical tropopause inversion layer using high-resolution temperature profiles retrieved from COSMIC GNSS Radio Occultation" by Noersomadi Noersomadi et al.:

This study investigated the characteristics of the tropopause inversion layer using the high vertical resolution GNSS-RO data. It gave more details of the TIL sharpness

and depth compared to previous studies, including the climatological mean and the intraseasonal to interannual variations. In particular, this work gives a special focus on two different longitude regions (the Maritime Continent and the Pacific Ocean). It extended previous work by related the interannual and intra-seasonal variations of the TIL to ENSO and MJO, respectively. It is well laid-out and well written. However, there are still some problems need to be fixed before publication. We appreciate the reviewer for providing constructive comments to our manuscript. We show below our responses to the individual comments.

Introduction The most relevant previous work to my understanding is Grise et al. 2010. Grise et al. 2010 already presented a global survey of the TIL strength (including annual cycle, horizontal distribution and interannual variations related to QBO) using the GPS-RO data with a vertical resolution also 0.1 km. However, it is not introduced in the introduction. It would be necessary to introduce results from Grise et al. 2010 and clearly describe what kind of improvement does this study want to have compared to them. We add the following statements in P2 L12-16: "Using routine radiosonde sounding data, a strong mean inversion at the tropopause in the midlatitude was analyzed by Birner et al. (2002) and Birner, (2006). Grise et al. (2010) conducted a global survey of the TIL characteristics, including annual cycle, horizontal distribution and interannual variations related to the stratospheric Quasi Biennial Oscillation (QBO) using the Global Positioning System Radio Occultation (GPS-RO) data."

We also add the statements citing Birner (2006) and Grise et al. (2010) in P2 L23-26. "Birner (2006) found that the mean N2 shows enhanced values near the extratropical tropopause compared to the extratropical lower stratosphere. Furthermore, Grise et al. (2010) found that the largest magnitudes of N2 found between $10°-15°$ latitude in both hemisphere during northern hemisphere (NH) winter season."

We add the following sentences in Section 3 (P6 L23-26): "We found the maxN2 at range $11-12.0 \times 10-4$ s$-2$ above CPT height in the two longitude regions. Note that Grise et al. (2010) reported that the zonal mean maxN2 value above the LRT and CPT

heights was $\sim 8.0 \times 10{-}4$ s${-}2$ using CHAMP dataset (see fig.2 in Grise et al, 2010). The different values found in this study are probably due to the use of data of higher effective vertical resolution."

We also add the following sentences in P6 L30-34: "The results shown in Fig. 3 uncover the detail structure of N2 above CPT in the specific longitude regions compared to the results by Grise et al. (2010) that showed the mean N2 over $0{-}1$ km layer above LRT. The vertical propagation of equatorial waves (Kelvin waves and/or gravity waves), as the results of convective forcing, modulates the tropopause (Tsuda et al., 1994; Randel and Wu, 2005; Kim and Alexander, 2015; Kim et al., 2018). The MJO activity was also found to control the tropopause variability (Kim and Son, 2012; Pilch Kedzierski et al., 2016)."

P2L12: "A very low temperature in the TIL", TTL should be better here

We follow your suggestion in P2 L18.

P3L6-10: I think it is now commonly known that the vertical resolution of the GPS/GNSS-RO data is up to 0.1 km. This sentence misleads the authors with a impression that the vertical resolution of GPS/GNSS-RO data is 1 km. Please rephrase this paragraph here. There are two fundamental retrievals of GPS-RO data called the wave optics (WO) and the geometrical optics (GO). We have mentioned the definitions of both the WO and GO in Section 2.1. The vertical resolution by GO is limited about 1-2 km (Kursinski et al., 1997), while the effective vertical resolution by WO or radio holographic method up to 0.1 km (Gorbunov, 2002). We used GPS-RO retrieved using WO (FSI) up to 30 km in the present study (Tsuda et al., 2011; Noersomadi and Tsuda, 2017). We think the reviewer referred to the wetPrf products provided by CDAAC. It is true the wetPrf products provide 0.1 km grid resolution, but the effective vertical resolution at 10-20 km altitudes of cosmic2013 is 0.5 km. For more details, please see the improved atmospheric data inversion called NEWROAM in the CDAAC webpage (https://cdaac-www.cosmic.ucar.edu/cdaac/doc/overview.html).

We add the following statements in P3 L28-30: The actual effective vertical resolution of RO measurements increases in regions of increased refractive gradients such as inversion layers at the top of the boundary layer or the tropopause.

Data P3L35-40: To my experience, the cosmic data from CDAAC should has data available for 0.1 km vertical resolution, e.g. wetPrf. Also note, most of studies as I know (Randel et al. 2007; 2010; Grise et al., 2010; Kedzierski et al., 2016 etc.) using cosmic data from the CDAAC and has vertical resolution of 0.1 km. I am not sure whether it is true that the cosmic2013 smoothed the data over a 0.5 km scale. Please check that carefully. Noersomadi and Tsuda (2017) investigated the three GPS-RO products, two by UCAR (i.e. COSMIC version 2010 and re-processed data version 2013) and one by RISH (hereafter we refer to cosmicfsi in the present study). Noersomadi and Tsuda (2017) showed that COSMIC version 2010 indicated mixture of vertical resolution (0.1 – 1 km), while COSMIC version 2013 fixed the resolution at 0.5 km in the UTLS (see their result in fig. 3). Detailed explanation about the retrieval of COSMIC re-processed data version 2013 can be found in Zeng et al (2016). Randel et al. (2007, 2010), Grise et al. (2010), and Kedzierski et al. (2016) did not mention the specific data version/product that they used. In this study, we used the cosmicfsi data product whose vertical resolution is 0.1 km near the tropopause (Noersomadi and Tsuda, 2017).

Figure 1: For the cosmic2013 data. The GPS-RO data is well known to be very accurate with high vertical resolution. From the Figure shown in Figure 1, the temperature profile is heavily smoothed. I strongly doubt for the results shown here. Please check carefully whether you are using the correct type of product from the cosmic2013 data.

Figure R1 Comparison of the T profiles on December 17, 2011, between the radiosonde at Jakarta Indonesia (black line) and nearby COSMIC GPS-RO by cosmicfsi (blue line), atmPrf cosmic2013 (red line), and werPrf cosmic2013 (green line), respectively.

Figure R1 is a revised version of fig. 3 of Noersomadi and Tsuda (2017) by changing the atmPrf version 2010 profile to wetPrf version 2013. Figure R1 shows that

both atmPrf and wetPrf of cosmic2013 indicate smoothed temperature profiles near the tropopause compared to our cosmicfsi data product. The wetPrf data is 0.1 km gridded data, but the actual, effective resolution is 0.5 km as described by Sokolovskiy et al. (2014) and Zeng et al. (2016).

TIL definition: There has been a lot of definitions to TIL strength, sharpness or thickness. Beside the authors mentioned, please also include Randel et al. 2007; 2010 and Wang et al. 2013 for the TIL definition using the temperature gradient. Randel,W. J.,Wu, F., and Forster, P.: The Extratropical Tropopause Inversion Layer: Global Observations with GPS Data, and a Radiative Forcing Mechanism, J. Atmos. Sci., 64, 4489, doi:10.1175/2007JAS2412.1, 2007. Randel, W. J. and Wu, F.: The Polar Summer Tropopause Inversion Layer, J. Atmos. Sci., 67, 2572–2581, doi:10.1175/2010JAS3430.1, 2010. Wang,W., Matthes, K., Schmidt, T., and Neef, L.: Recent variability of the tropical tropopause inversion layer, Geophys. Res. Lett., 40, 6308–6313, doi:10.1002/2013GL058350, 2013. We add the following statements in P5 L13-17: "Grise et al. (2010) showed minor differences in the zonal mean N2 profile relative to LRT and CPT in the tropics. Randel et al. (2007; 2010) and Wang et al. (2013) investigated the TIL using the temperature gradients. Schmidt et al. (2005) and Son et al. (2011) defined the TIL with respect to the LRT height; more recently, Gettelman and Wang (2015) and Pilch Kedzierski et al. (2016) used the definition with respect to LRT height."

Definition of the TIL thickness: I don't know whether it is necessary to have new definitions of TIL sharpness and thickness since there have been so many kinds of definitions. If the authors feel it is necessary, please address the reason clearly here. In particular, why a 80% is used for the dH. Are there any physical or statistical reasons? Otherwise, I would suggest to using the existed definitions, for example, the maximum of N2 above the tropopause for the TIL sharpness. We add the following sentences in P5 L33-37: "We will focus on seasonal variation of S-ab and dH in Section 3.3. We have analyzed how the results change quantitatively when using the difference between maxN2+1 and minN2−1 instead of averaging along ±1 km relative to CPT as was done by Kim and Son (2012), with effectively higher vertical resolution dataset. In order to obtain dH we need to define the corresponding N2 value. Considering the stable N2 value in the lower stratosphere as 6.5 ×10−4 s−2 and the maxN2 as 10.5 ×10−4 s−2 (Fig.2 right panel), the threshold of N2 should be larger than 65% of maxN2. We choose 80% as the threshold."

Results: For the global distribution and seasonal variations of the N2 and TIL, it would be very helpful to have some comparison to previous work like Grise et al. 2010. Please describe clearly whether the results are consistent with each other and what are new findings from the previous work. We have modified the structure of the manuscript focusing on the N2 relative to CPT.

Please see replies to the first comment by the reviewer.

Regarding the comparison with previous studies, we add the following statements in P7 L41 – P8 L6: "Large S-ab values are found along the equatorial region, while low OLR regions show latitudinal variation with season. Local and seasonal variability of horizontal structure of tropopause sharpness presented in this work is consistent with previous studies which attributed it to equatorial waves activity (e.g. Grise et al., 2010; Son et al., 2011; Kim and Son, 2012). However, we found different quantitative results in particular over the Western Pacific because of our use of maxN2+1 and minN2−1 instead of averaging N2 within ±1 km relative to CPT by Kim and Son (2012), and also because of the use of data of higher effective vertical resolution. Maximum static stability just above the tropical tropopause could also be associated with divergence flow as demonstrated by Pilch Kedzierski et al. (2016)."

Figures 5 and 7. The values of N2 above the tropopause is not clear. Please update the color map used for these figures. We have revised the manuscript by removing some figures (including Figs. 5 and 7 of the original manuscript) and updating the figures number. We also update the color map of the Fig.3, 4, 6.

Thank you very much again for your very valuable comments and suggestions.

References: Birner, T., Dörnbrack, A., and Schumann, U.: How sharp is the tropopause at midlatitudes?, Geophys. Res. Lett., 29, 1700, doi:10.1029/2002GL015142, 2002. Birner, T.: Fine-scale structure of the extratropical tropopause region, J. Geophys. Res., 111, D04104, doi:10.1029/2005JD006301, 2006. Gettelman, A., and Wang, T.: Structural diagnostics of the tropopause inversion layer and its evolution, J. Geophys. Res., 120, 46–62, doi:10.1002/2014JD021846, 2015. Grise, K. M., Thompson, D. W. J., and Birner, T.: A global survey of static stability in the stratosphere and upper troposphere, J. Clim., 23, 2275–2292, doi:10.1175/2009JCLI3369.1, 2010. Gorbunov, M. E.: Radio-holographic analysis of Microlab-1 radio occultation data in the lower troposphere, J. Geophys. Res., 107, 4156, doi:10.1029/2001JD000889, 2002. Kedzierski, R. P., Matthes, K. and Bumke, K.: The tropical tropopause inversion layer: Variability and modulation by equatorial waves, Atmos. Chem. Phys., 16, 11617–11633, doi:10.5194/acp-16-11617-2016, 2016. Kim, J.-E., and Alexander, M. J.: Direct impacts of waves on tropical cold point tropopause temperature, Geophys. Res. Lett., 42, 1584–1592, doi:10.1002/2014GL062737, 2015. Kim, J. and Son, S. -W.: Tropical cold-point tropopause: Climatology, seasonal cycle, and intraseasonal variability derived from COSMIC GPS radio occultation measurements, J. Clim., 25, 5343–5360, doi:10.1175/JCLI-D-11-00554.1, 2012. Kursinski, E. R., Hajj, G. A., Schofield, J. T., Linfield, R. P. and Hardy, K. R.: Observing Earth's atmosphere with radio occultation measurements using the Global Positioning System, J. Geophys. Res., 102(D19), 23,429-23,465, doi:10.1029/97JD01569, 1997. Noersomadi and Tsuda, T.: Comparison of three retrievals of COSMIC GPS radio occultation results in the tropical upper troposphere and lower stratosphere, Earth Planets Space., 69, doi:10.1186/s40623-017-0710-7, 2017. Randel, W. J., & Wu, F.: Kelvin wave variability near the equatorial tropopause observed in GPS radio occultation measurements, Journal of Geophysical Research, 110, D03102. https://doi.org/10.1029/2004JD005006, 2005. Sokolovskiy, S., Schreiner, W., Zeng, Z., Hunt, D., Kuo, Y. -H, Meehan, T. K., Stecheson, T.W., Manucci, A.J., Ao, C.O.: Use of the L2C signal for inversions of GPS radio occultation

data in the neutral atmosphere, GPS Solut, 18, 404−416, doi:10.1007/s10291-013-0340-x, 2014. Tsuda, T., Murayama, Y., Wiryosumarto, H., Harijono, S. W. B., and Kato, S.: Radiosonde observations of equatorial atmosphere dynamics over Indonesia: 1. Equatorial waves and diurnal tides, J. Geophys. Res., 99, 10491–10505, doi:10.1029/94JD00355, 1994. Zeng, Z., Sokolovskiy, S., Schreiner, W., Hunt, D., Lin, J. and Kuo, Y. H.: Ionospheric correction of GPS radio occultation data in the troposphere, Atmos. Meas. Tech., 9, 335–346, doi:10.5194/amt-9-335-2016, 2016.

Please also note the supplement to this comment:
https://www.atmos-chem-phys-discuss.net/acp-2018-1182/acp-2018-1182-AC2-supplement.pdf

[Figure]

Temperature (K)

**Fig. 1.** Comparison of the T profles on December 17, 2011, between the radiosonde at Jakarta Indonesia (black line) and nearby COSMIC GPS-RO by cosmicfsi (blue line), atmPrf cosmic2013 (red line), and werPrf c

---

## Author Comment (AC3) · 25 Mar 2019

Reply to the comments by the Reviewer #3

This manuscript presents global characteristics of static stability in different vertical co-ordinate systems and closely investigates its characteristics in the tropical tropopause region. The authors use ten years (2007 to 2016) of GNSS RO data from the COSMIC satellite constellation. Data from an FSI RO retrieval are used. These profiles have a much better vertical resolution than profiles from other RO processing centers. The

tremendous vertical resolution of this data set is exploited by investigating atmospheric characteristics in detail. Spatial, inter-annual, annual, and intra-interannual variability of tropopause sharpness and the thickness of the tropopause inversion layer (TIL) are discussed. In my opinion, this is an interesting study with important scientific results. To a large extend, the manuscript is clearly and concisely written. Exceptions are some parts of the introduction and data description (see minor comment below). More importantly, I have some doubts regarding the vertical resolution of the data set and the conclusions drawn (see major comment below).

We appreciate the reviewer for providing constructive comments to our manuscript. We show below our responses to the individual comments.

1 General comments

1. The vertical resolution of the data set is specified to be 100 m. However, in some sections of the manuscript (mainly in section 3.3), the authors discuss characteristics with a distinctively better vertical resolution. Middle panels of Fig. 10 and also Fig. 13, e.g., show dH characteristics with a 10 m vertical resolution. I doubt somehow that this appropriate. Similarly, the authors find an uncertainty of the thickness of the TIL of 40 m. It is important to add a discussion and prove that these features can really be retrieved with this data set. See also minor comments on the retrieval and the uncertainty of the vertical grid. The vertical resolution of the dataset (i.e. 100 m) can be regarded as the uncertainty of each data point. Averaging a lot of data points with an uncertainty (100 m) would reduce the uncertainty for the average. In other words, this is similar to the standard error of the mean. The uncertainty of the thickness of TIL (40 m) is statistically reasonable. We used 80% of maxN2 within 1 km above CPT. To find the point where N2 at 80% reaching to and decreasing from maxN2, we applied a linear interpolation between the nearest two points within 100 m (Fig. R1). Therefore, it is reasonable that the random values of dH have the uncertainty of the order 10 m (the colormap legend in Fig.6 middle row range between $0.39-0.48$ km). Then, we average dH from many profiles.

Figure R1 The diagram of the algorithm to calculate dH from the N2 profile.

2 Minor comments

• I have difficulties in following the logical structure of some parts of the introduction (section 1) and data description (section 2). More specifically, paragraphs 3, 4, and 5 of section 1 (pages 2 and 3), and several paragraphs of section 2 (i.e., the first paragraph of section 2, 2.1, and 2.2, the last paragraph of section 2.1 (concerning the TIL definition)) should be revised. We have revised the logical structure of the introduction. We refer to the Fig. 2 (right panel) for the clear description in P3 L6-9.

• Add some more detailed discussion about the data retrieval. What input data are used for the FSI retrieval? What is the vertical coordinate of your data set (height above ellipsoid, height above geoid, geopotential height)? What is the uncertainty of the vertical grid? Scherllin-Pirscher et al. (2017, doi:10.1002/2016JD025902) also discuss some of these issues. We used the same retrieval provided by CDAAC/UCAR. We modified the sewing height between FSI and GO retrieval to $\sim$30 km to obtain better vertical resolution in the UTLS (Tsuda et al., 2011; Noersomadi and Tsuda, 2017). The input data is atmPhs (atmospheric excess phase) provided by CDAAC.

We have mentioned in the manuscript that we used geopotential height (P4 L39). "We adjusted T from GPS-RO in the geometrical height domain to the geopotential height used for radiosonde data before performing the comparison."

We add the following sentence (P4 L23-24): "The discrepancy in the CPT altitude between cosmicfsi and the campaign radiosonde dataset was 70 m (mean) and 100 m (median) (Noersomadi and Tsuda, 2017)."

• Introduce all acronyms at their first occurrence. We follow this suggestion.

• Abstract, line 22: It is not clear at this point what "S-ab anomaly (S-ab?)" and "OLR anomaly" refer to. Please rewrite this sentence.

We define the S-ab and OLR anomaly in the abstract.

• Introduction, page 2, paragraph 3: At least for a non-expert reader, the first two sentences "The vertical profile of N2 across the tropopause (i.e., the sharpness) and the thickness of the layer of maximum N2 above the tropopause have been determined in previous studies using both ground- and satellite-based observations. For example, Bell and Geller (2008) analyzed the twice daily standard radiosonde data from the WMO stations and found that the thickness was âĹij1 km at low latitudes." are not clear. Does "the thickness" refer to the layer between the tropopause and maximum N2 above the tropopause? Please clarify. We explicitly write as "the thickness of the layer of maximum N2 above the tropopause (i.e., between 80% reaching to and decreasing from maximum N2)". Also, we refer to the Fig. 2 (left panel) for the clear description (P3 L6-9).

• Introduction, page 2, paragraph 4: the better reference for the COSMIC data set is Anthes et al. (2008, doi:10.1175.BAMS-89-3-313). We follow this suggestion (P3 L16).

• Introduction, page 3, paragraph 2: Add the Noersomadi and Tsuda (2017)- reference. We follow this suggestion (P3 L31).

• Section 2.1, page 3, line 27: As far as I know, COSMIC does not provide 1500 to 2000 profiles anymore. Please clarify. We add the following statement in P4 L10-11: "Nevertheless, the number of profiles is significantly decreasing in 2015 and 2016 $(60-100$ profiles over $10°S–10°N$)."

• Section 2.1 page 4, lines 6/7: I do not understand the explanation "caused by different truncations of the GNSS signals in the lower atmosphere". It is true that the penetration depth is different for each RO measurement. It is also true that the penetration depths of the retrieved profiles can be different for different RO retrievals. However, I do not understand the connection between penetration depth and the number of profiles. Is there a specific quality indicator which reduces the number of profiles for cosmicfsi? Please clarify. We have slightly modified the atmospheric data inversion

called ROAM provided by CDAAC and do not understand details very much. We believe CDAAC used improved atmospheric data inversion (NEWROAM) as mentioned in their webpage (https://cdaac-www.cosmic.ucar.edu/cdaac/doc/overview.html). One possibility is due to different retrieval algorithm and background extrapolation model for L1 and L2 bending angles. We retrieved cosmicfsi using FSI algorithm, while cosmic2013 were re-processed using Phase Matching algorithm (Sokolovskiy et al., 2014; Zeng et al., 2016; Sokolovskiy, personal discussion, 2017). However, discussion on the differences in the retrieval algorithm between cosmicfsi and cosmic2013 is beyond the scope of the present study.

We add the following sentence in P4 L30-32: "The difference in the total number of occultations in the two retrievals is possibly caused by different algorithm and background extrapolation model for bending angles as reported by Noersomadi and Tsuda (2017). Discussion of difference between retrieval algorithm for cosmicfsi and for cosmic2013 is beyond the scope of this study."

• Section 2.1, page 4, lines 15/16: "which is located within 115 km horizontal radius": Does this number refer to the mean tangent point location? How is it defined? Did you account for the tangent point drift? Please explain. We refer to the distance between the perigee point (tangent point) and the location of radiosonde launch site. We defined the radius as the distance between radiosonde station and the perigee point. We did not account for the tangent point drift. The cosmicfsi and cosmic2013 data show differences of ±0.4 K and less than 100 m for CPT temperature and CPT height, respectively, within the distance of 200 km and the time difference of ±3 hours (Noersomadi and Tsuda, 2017).

• Section 2.2, page 4, line 36 to page 5, line 5: I recommend referring to the right panel of Fig. 2 for this explanation. Furthermore, the figure should include all parameters introduced in this section. We have referred to Fig. 2 (right panel) in the Introduction for the clear description (P3 L6-9). Definition of all TIL parameters are summarized in Table 2. Since we focus on S-ab and dH in the later discussion, we

emphasize only these parameters in Fig. 2 (right panel). We are afraid that the figure would be too complicated if we included all the parameters. Thus, we would like to keep it as it is.

• Section 3.1, page 5, lines 33/34: "Figure 3a-d...": This sentence is a general statement, which should be made earlier in this paragraph. This figure has been removed in the revised manuscript as suggested by other reviewer.

• Section 3.1, page 6, line 1: "the LRT should be the same as the lowest CPT" and line 40: "the LRT and CPT are at the same location/height". In the tropics, LRT is usually lower than CPT because LRT refers to a specific temperature gradient and CPT to the temperature minimum. Please clarify. We have shortened the text in Section 3.1 to focus on discussing the mean N2 relative to CPT height in the tropics. Thus, these sentences have been removed.

• Section 3.1, page 6, lines 39 to 42: I do not understand this explanation. Please rewrite. We have modified this paragraph (P6 L27-33) as follows. "The profiles of large N2 over 20°N and 20°S in the MC region represent the vertical section of the Kelvin-wave and mixed Rossby–gravity-wave response known as the Matsuno–Gill pattern mode (Matsuno, 1966; Gill, 1980; Grise et al., 2010; Nishimoto and Shiotani, 2012). The results shown in Fig. 3 uncover the detailed structure of N2 above the CPT in the specific longitude regions, being compared to the results by Grise et al. (2010) who showed the mean N2 over $0-1$ km layer above the LRT. The vertical propagation of equatorial waves (i.e., Kelvin waves and/or gravity waves), as the results of convective forcing, modulates the tropopause (Tsuda et al., 1994; Randel and Wu, 2005; Kim and Alexander, 2015; Kim et al., 2018). The MJO activity was also found to control the tropopause variability (Kim and Son, 2012; Pilch Kedzierski et al., 2016)."

• Section 3.1, page 7, lines 15/16: I do not understand the sentence "The level of increasing temperature can be determined as the LRT height when the WMO definition is attained" as the LRT is not determined by increasing temperatures but only a

temperature gradient threshold. We have shortened the text in Section 3.1 to focus on discussing the mean N2 relative to CPT height in the tropics. Thus, this sentences was removed.

• Section 3.2, page 7, Figure 9: The class widths of several panels of Fig. 9 seem to be either non-equidistant or floating numbers causing rounding errors. Please use well defined and equidistant classes. We have updated Fig.9 and change the figure number into Fig.5 in the revised manuscript.

• Section 3.2, page 7, lines 28/29: Since Fig. 9 shows that mean S-aCPT is smaller than 6:6 $\times$ 10-4 s-2 this statement cannot be true. Please clarify. The mean S-aCPT is 6.4 $\times 10-4$ s$-2$. We modify the sentence (P7 L12) as follows: The values of S-aCPT are mostly in the range $2.8-6.6 \times 10-4$ s$-2$ ...

• Section 3.3, page 8, lines 13/14: "The highest values, up to 16 - 18 $\times$ 10-4 s-2, are associated with low OLR values...": In DJF, S-ab is clearly above average (17 - 18 $\times$ 10-4 s-2) west of South America, where OLR values are about 260 W/m2, defined as "non-convective" on page 8, lines 33/34. Above the convective regions in South America, however, where OLR values are really low, the S-ab only reaches about 14 $\times$ 10-4 s-2. So this statement seems to be wrong for the South American region. Please clarify. We add the following sentences in P7 L41 – P8 L9: "Large S-ab values are found along the equator region, while low OLR regions show latitudinal variation with season. Local and seasonal variability of horizontal structure of tropopause sharpness presented in this work is consistent with previous studies which attributed it to equatorial waves activity (e.g. Grise et al., 2010; Son et al., 2011; Kim and Son, 2012). However, we found different quantitative result in particular over the Western Pacific because using maxN2+1 and minN2–1 instead averaging N2 within $\pm$1 km relative to CPT by Kim and Son (2012), and also because of the use of data of higher effective vertical resolution. Maximum static stability just above the tropical tropopause could also be associated with divergence flow as demonstrated by Pilch Kedzierski et al., (2016). Large S–ab around the 240°–270°E longitude region in DJF is qualitatively related to OLR values

of 220–240 W m–2 representing the inter tropical convergence zone (ITCZ). Large S-ab values of 14–16 × 10−4 s−2 are also found over South Asia and near the ITCZ in JJA. Exception is seen over South America where S-ab only reaches ∼14 × 10−4 s−2 associated with OLR values < 220 W m–2."

• Section 3.4, page 9, line 22: How big are the cross-correlations between S-ab and OLR at different lags? We add the following sentence in P9 L13-15. "We have tested for different lags. The cross-correlation between S-ab and OLR become smaller for two month lag (−0.49 and −0.57 over MC and PO regions, respectively)."

3 Figures

We have updated the figures following all the suggestions below by the reviewer, including color map updates, grid line additions, and legend revisions. Note there are also some other comments above.

• I recommend adding a background grid in Figs. 1, 2, 11, 12, 13, 14, 16, 18 • Please also add minor ticks on x-axis of Figs. 11, 14, right panel of Fig. 18, and on y-axis of Figs. 12, 13, 16

• I recommend adding "MC" and "PO" next to the longitudinal information in the figure titles of Figs. 3, 5, 6, 7, 14, 15, 16.

• Please indicate MC, PO, Atlantic, and Indian Ocean in Figs. 4, 7, and 8 (e.g., arrows below the x-axis).

• Indicate MC and PO in the figure legends of Figs. 12, 13 (could be one two panel figure) and also in the right panel of Fig. 16.

• I suggest indicating the LRT and CPT for all data sets in Fig. 1

• Indicate El Nino events 2009/2010 and 2015/2016 in both panels of Figs. 14 and 15.

4 Editorial We have edited following all the suggestions below by the reviewer.

• Page 1, line 15: "in the range" ! "in the range of"

• Page 3, line 8: "Centre" ! "Center"

• Page 3, line 17: "cosmic2013 only" ! "cosmic2013 data only"

• Page 4, line 5: "within a 183-day period" ! "within the 183-day period in 2011/2012".

• Page 4, line 16: According to Table 1, Surabaya station is located at 112.78◦E, 7.37◦S. Please clarify.

• Page 4, line 18: "less than 30 min": This is only true for the UTLS.

• Page 4, line 21: "within a 183-day period" ! "within the 183-day period".

• Page 6, line 4: "height-longitude" ! "longitude-height"

• Page 6, line 10: "height-latitude" ! "latitude-height"

• Page 6, line 30: "Fig. 6b" ! "Fig. 6c"

• Page 7, line 4: "height-longitude" ! "longitude-height"

• Page 7, line 19: "than in JJA." ! "than in JJA (Fig. 8)".

• Page 7, line 27: "entire latitude range" ! "entire region from"

• Page 10, line 27: "15 April 2012" might be "13 April 2012" Fig. 12 (right panel) does show the data until 15 April 2012.

Thank you very much again for your very valuable comments and suggestions.

References: Birner, T., Dörnbrack, A., and Schumann, U.: How sharp is the tropopause at midlatitudes?, Geophys. Res. Lett., 29, 1700, doi:10.1029/2002GL015142, 2002. Birner, T.: Fine-scale structure of the extratropical tropopause region, J. Geophys. Res., 111, D04104, doi:10.1029/2005JD006301, 2006. Gettelman, A., and Wang, T.: Structural diagnostics of the tropopause inversion layer and its evolution, J. Geophys. Res., 120, 46–62, doi:10.1002/2014JD021846, 2015. Gill, A. E.: Some

simple solutions for heat-induced tropical circulation, Q. J. Roy. Meteorol. Soc., 106, 447–462, 1980. Grise, K. M., Thompson, D. W. J., and Birner, T.: A global survey of static stability in the stratosphere and upper troposphere, J. Clim., 23, 2275–2292, doi:10.1175/2009JCLl3369.1, 2010. Kedzierski, R. P., Matthes, K. and Bumke, K.: The tropical tropopause inversion layer: Variability and modulation by equatorial waves, Atmos. Chem. Phys., 16, 11617–11633, doi:10.5194/acp-16-11617-2016, 2016. Kim, J.-E., and Alexander, M. J.: Direct impacts of waves on tropical cold point tropopause temperature, Geophys. Res. Lett., 42, 1584–1592, doi:10.1002/2014GL062737, 2015. Kim, J. and Son, S. -W.: Tropical cold-point tropopause: Climatology, seasonal cycle, and intraseasonal variability derived from COSMIC GPS radio occultation measure-ments, J. Clim., 25, 5343–5360, doi:10.1175/JCLI-D-11-00554.1, 2012. Matsuno, T. : Quasi-geostrophic motions in the equatorial area, J. Meteorol. Soc. Jpn., 44, 25–42, 1966. Noersomadi and Tsuda, T.: Comparison of three retrievals of COSMIC GPS radio occultation results in the tropical upper troposphere and lower stratosphere, Earth Planets Space., 69, doi:10.1186/s40623-017-0710-7, 2017. Randel, W. J., & Wu, F.: Kelvin wave variability near the equatorial tropopause observed in GPS radio occultation measurements, Journal of Geophysical Research, 110, D03102. https://doi.org/10.1029/2004JD005006, 2005. Sokolovskiy, S., Schreiner, W., Zeng, Z., Hunt, D., Kuo, Y. -H, Meehan, T. K., Stecheson, T.W., Manucci, A.J., Ao, C.O.: Use of the L2C signal for inversions of GPS radio occultation data in the neutral atmosphere, GPS Solut, 18, 404−416, doi:10.1007/s10291-013-0340-x, 2014. Tsuda, T., Murayama, Y., Wiryosumarto, H., Harijono, S. W. B., and Kato, S.: Radiosonde observations of equatorial atmosphere dynamics over Indonesia: 1. Equatorial waves and diurnal tides, J. Geophys. Res., 99, 10491–10505, doi:10.1029/94JD00355, 1994. Tsuda, T., Lin, X., Hayashi, H., and Noersomadi: Analysis of vertical wave number spectrum of atmospheric gravity waves in the stratosphere using COSMIC GPS radio occultation data, Atmos. Meas. Tech., 4, doi:10.5194/amt-4-1627-2011, 2011. Zeng, Z., Sokolovskiy, S., Schreiner, W., Hunt, D., Lin, J. and Kuo, Y. H.: Ionospheric correction of GPS radio occultation data in the troposphere, Atmos. Meas. Tech., 9,

335–346, doi:10.5194/amt-9-335-2016, 2016.

Please also note the supplement to this comment:
https://www.atmos-chem-phys-discuss.net/acp-2018-1182/acp-2018-1182-AC3-supplement.pdf

[Figure]

[Figure]

The vertical distance between two nearest blue circles is 100 m (i.e. vertical grid resolution)

dH

This point could be 70-80% of maxN$^2$

This point could be 80-90% of maxN$^2$

maxN$^2$

This point could be 80-90% of maxN$^2$

This point could be 70-80% of maxN$^2$

**Fig. 1.** The diagram of the algorithm to calculate dH from the N2 profile.

---

## Author Response (AR2)

**Co-Editor Decision: Publish subject to technical corrections** (17 Apr 2019) by Peter Haynes

Comments to the Author:

All referees are in agreement that your paper is now suitable for publication in ACP. One referee has recommended some technical corrections -- please can you consider these recommended corrections carefully (and perhaps also check the paper more generally) before providing a final version.
* * *
**1**

Review of "Influence of ENSO and MJO on the zonal structure of tropical tropopause inversion layer using high-resolution temperature profiles retrieved from COSMIC GPS Radio Occultation" by Noersomadi et al.

The manuscript has improved a lot upon review, the authors have satisfied all of my comments, it reads very well overall with the right amount of detail and discussion in each section. It's a high-quality study and I recommend to publish it after a few very minor technical corrections:

We deeply appreciate the reviewer for accepting the manuscript to be published in ACP. We show below our responses to the individual corrections.

**p.8 l.34: 'convective activity resulted in decreased...' --> I suggest a more correct expression 'regions with more frequent convective activity show decreased...' because strictly speaking these are climatologies.**

We follow suggestion by the reviewer and revise the statement in P8 L34-35.

**p.9 l.25-29: During El Nino events convection moves away from the MC towards the PO region and the eastern Pacific, so it's to be expected that the correlations with the ENSO index will be opposite for MC and PO. The high magnitude of the correlations for PO could be due to the fact that, outside ENSO events, this is a region with very low convective activity, whereas MC is not devoid of convection even during ENSO events**

(and MJO can influence the MC region as well in its late phases), meaning more processes will affect OLR and Sab there.

--> I suggest to slightly rephrase this part of the paragraph to include the points from above.

We agree with the reviewer and add the following sentence in P9 L27-30:

The high magnitude of the correlation for PO could be due to the fact that, outside ENSO events, this is a region with very low convective activity, whereas MC is not devoid of convection even during ENSO events (and MJO can influence the MC region as well in its late phases).

**p.11 l.5-9: This mechanism can also explain the differences in climatology between MC and PO as well as interannual to intraseasonal variability, since ENSO and MJO modulate convective activity. You test this with Fig. 14.**

**p.11 l.10-17: you can link your result from Fig. 14 to the modulation of convection by ENSO and MJO from previous sections. It's really worth highlighting that your Fig. 14 kind of summarizes the forcing of tropical TIL variability across time-scales.**

We appreciate for suggestions by the reviewer and add the link to Fig.12 and Fig. 10 in P11 L14.

--> Also, I noticed sometimes there's a space and/or capital letter missing when you refer to figures (Fig.X / fig.X --> Fig. X) please check throughout the manuscript for consistency

Thank you very much for your corrections.

####
**Concluding remarks:**

- Refer to the corresponding figure number when you introduce your results.

- Highlight what's new. Generally in the first two paragraphs your results agree very well with previous studies (refer to them as well), while your dataset has improved vertical resolution compared to them (this should be highlighted); and the last two paragraphs show variability in TIL properties (ENSO and MJO) that have not been studied in this detail previously.

We follow suggestions by the reviewer and add the link to figures in P11 L25, L27, L33, L36, and P12 L1, L8, L12.

We add the statements in P11 L27-29:

The results generally agree very well with previous studies (Birner et al., 2002; Grise et al., 2010). Using dataset with improved vertical resolution, we found larger values of $N^2$ compared to the results by Grise et al (2010) in this study.

We also add the sentece in P12 L9-10:

We showed influence of ENSO and MJO on the variation of TIL that has not been studied previously.

**Fig. 8: you may want to check the legend labelling (PO should be solid red)**

We have edited the legend of Fig. 8.

Thank you very much again for your very valuable comments and suggestions.